# Application of *Caenorhabditis elegans* in Lipid Metabolism Research

**DOI:** 10.3390/ijms24021173

**Published:** 2023-01-07

**Authors:** Lu An, Xueqi Fu, Jing Chen, Junfeng Ma

**Affiliations:** National Engineering Laboratory for AIDS Vaccine, School of Life Sciences, Jilin University, Changchun 130012, China

**Keywords:** *Caenorhabditis elegans*, obesity, lipid metabolism, fat-regulatory pathways, fat storage characterization

## Abstract

Over the last decade, the development and prevalence of obesity have posed a serious public health risk, which has prompted studies on the regulation of adiposity. With the ease of genetic manipulation, the diversity of the methods for characterizing body fat levels, and the observability of feeding behavior, *Caenorhabditis elegans* (*C. elegans*) is considered an excellent model for exploring energy homeostasis and the regulation of the cellular fat storage. In addition, the homology with mammals in the genes related to the lipid metabolism allows many aspects of lipid modulation by the regulators of the central nervous system to be conserved in this ideal model organism. In recent years, as the complex network of genes that maintain an energy balance has been gradually expanded and refined, the regulatory mechanisms of lipid storage have become clearer. Furthermore, the development of methods and devices to assess the lipid levels has become a powerful tool for studies in lipid droplet biology and the regulation of the nematode lipid metabolism. Herein, based on the rapid progress of *C. elegans* lipid metabolism-related studies, this review outlined the lipid metabolic processes, the major signaling pathways of fat storage regulation, and the primary experimental methods to assess the lipid content in nematodes. Therefore, this model system holds great promise for facilitating the understanding, management, and therapies of human obesity and other metabolism-related diseases.

## 1. Introduction

With the improvement in living standards, most people are more inclined to have high-energy diets and sedentary lifestyles, which greatly increases the risk of inducing obesity. As a result, the global obesity epidemic is prevalent and worsening in most parts of the world. To date, the obese population has exceeded 650 million all over the world and is still increasing with a younger trend [1]. Obesity is a chronic disease resulting from an abnormal metabolism, characterized by an excessive fat accumulation and may further cause type Ⅱ diabetes [2,3], cardiovascular diseases [4,5,6], non-alcoholic fatty liver disease [7], chronic kidney disease [8], hypertension, hyperlipidemia [9], sleep apnea, and tumors [10]. Accompanied by physical ailments, obese patients can also experience issues with mental health from receiving different visions. As a rule, chronic obesity-related disease not only causes the suffering of the organism but also imposes tremendous burdens on public health care and the social economy.

Consequently, it is crucial to maintain normal body fat levels in the organism. In this context, the clarification of the regulatory mechanisms involved in lipid storage and the identification of chemical substances regulating the fat metabolism will greatly contribute to maintaining an energy balance and controlling body fat. As more than 65% of the genes of *Caenorhabditis elegans* (*C. elegans*) are associated with humans [11], *C. elegans* is increasingly applied as a model in studies of lipid homeostasis. Over the last decade or so, studies on nematodes have helped immensely to reveal the ancient and complex mechanisms of fat regulation in humans. This review provides a brief overview of the lipid metabolic processes and the relevant genes and pathways that regulate lipid storage in *C. elegans*. In addition, several of the methods involved in the lipid distribution and content in *C. elegans* were also outlined.

This review brings an introduction and summary of the lipid-related content in nematodes for interested readers.

## 2. Advantages and Disadvantages of *C. elegans* as a Model for Studying Fat Metabolism

It is generally known that *C. elegans* has become a versatile and popular model for investigating the development and treatment of obesity and studying the mechanisms of the lipid metabolism. First, *C. elegans* is the first animal model whose genome is completely sequenced [12] and which has ~400 genes (over 70%) relating to the lipid metabolism in *C. elegans* which are homologous to mammals [13]. That is, many complex processes, including the food intake, fatty acid biosynthesis, transport, storage, and catabolism [14], are conserved in this simple organism. In addition, broad genetic and genomic tools, including targeted deletions, transgenics, tissue-specific RNA interference (RNAi) [15], mutagenesis screens, and a genome-scale RNAi screen [16], have been combined with rapidly quantifiable phenotypes for the fat content and nutrition intake to study the energy balance and fat metabolism [17]. Diverse mutants are available on the basis of wild-type nematodes because of their genetically tractable systems and the use of powerful genetic tools. For example, the CRISPR/Cas9 approach was employed in the lipid metabolism to generate several knockout strains to exclude the possibility of off-target effects. By gene knockout technology, HLH-11, encoded by a *helix loop helix-11* (*hlh-11*), was proved to regulate the lipid metabolism in response to the availability of food [18]. Moreover, as a member of the lipid-binding protein family, *lipid binding protein-5* (*lbp-5*) is predicted to be involved in the transport of fatty acids. To create *lbp-5* rescue worms, plasmid DNA containing the *lbp-5* full gene::GFP (green fluorescent protein) was injected into worms to construct the *lbp-5(tm1618)* mutant and then transgenic worms were selected by detecting the expression of GFP from the progeny [19].

Furthermore, compared to the traditional models for studying obesity, i.e., mice and rats, *C. elegans* has many advantages, such as a small size, simple structure (Figure 1), it is easy to cultivate, the hermaphrodites reproduce by self-fertilization, it has large numbers of progeny, it takes a simple measurement of the food intake and energy expenditure, has a short life span, and a low research cost. As the transparent body, it is easy to trace and quantify the lipids in *C. elegans* by using lipid-specific dyes and in vivo fluorescence markers. In addition, the use of *C. elegans* for experiments does not require the approval of the Institutional Animal Care and Use Committee [20]. All of these advantages make *C. elegans* a suitable and convenient in vivo model to identify obesity-promoting compounds or to assess the lipid-lowering capacity of natural products, which could facilitate the prevention of obesity and the identification of potential anti-hyperlipidemic drugs [21,22]. Additionally, as lipids play a vital role in the biology of nematodes, various analytical techniques, such as the visualization of lipid droplets and the analysis of the total fatty acids, have been frequently used to explore the lipid metabolism in *C. elegans*. Recently, as a new technique developed from metabolomics, lipidomics has also been gradually applied to analyze the lipids in nematodes comprehensively, which allows *C. elegans* to be more easily used for lipid studies. Thus, with conserved fat regulatory pathways and multiple methods of assessing the experimental alterations to the fat metabolism, *C. elegans* has been widely applied for exploring the lipid metabolism and its potential effects and mechanisms.

On the other hand, the ability of nematodes to synthesize a kind of ω-3 desaturase that is not present in most animals is undoubtedly another advantage for them in lipid metabolism studies. Meanwhile, due to the feasibility and simplicity of the operation, a fatty acid supplementation has become a common means of modifying the composition of the fatty acids in nematodes and is often used in experiments to rescue a fatty acid deficiency in mutants. Specifically, Qi et al. treated wild-type nematodes with the ω-3 fatty acid α-linolenic acid (ALA) and found a dose-dependent extension of the lifespan by altering their body fatty acid composition [23]. Similarly, a dietary supplementation with linoleic acid (LnA) and oleic acid (OA) could also contribute to the maintenance of longevity in wild-type nematodes [24]. Surprisingly, dietary fatty acids can also promote LD diversity through a seipin enrichment in an ER subdomain [25]. Cao et al. confirmed that the enrichment of SEIP-1 required PUFAs and microbial cyclopropane fatty acids (CFAs) through fatty acid supplement experiments [25]. In addition, a dietary supplementation of n-6 PUFAs can save the lipid-rich osmotic barrier defect in the innermost layer of *seip-1* homozygous mutant embryos [26]. Accordingly, fatty acids have been shown to be essential in the physiological regulation of *C. elegans* by taking advantage of the fact that they can be supplemented through diet.

In addition, the discovery of the association between the lipid metabolism and other physiological processes in *C. elegans* has also become its strength in lipid biology. In recent years, a number of studies have revealed that the lipid metabolism in *C. elegans* is also associated with pathogenesis, especially a bacterial infection, which is rarely seen in other model organisms. As bacterivores, the intestinal infections of *C. elegans* can be easily achieved by feeding. Hence, combined with the genetic manipulability of the genes related to the lipid metabolism, nematodes are particularly suitable for the in-depth excavation of the association between metabolism disorders and a pathogen infection. For instance, *C. elegans* deficient in the production of oleate are hypersusceptible to an infection with diverse human pathogens, which could be rescued by the addition of exogenous oleate [27]. This suggests that the synthesis of oleate is necessary for the nematodes’ immune activation and resistance to infection by various bacterial pathogens. Notably, the depletion of oleate in *skinhead-1* (*skn-1*) gain-of-function mutants’ nematodes led to the redistribution of somatic lipids in the previous study [28]. In-depth, redirected SKN-1 reduced the negative metabolic outcomes of a sensing pathogen infection in *C. elegans* [29]. Thus, it is clear that *C. elegans* can guide an intensive exploration of disorders of the lipid metabolism and their relationship to disease development as unique model organisms. In addition, *C. elegans* may also provide insights into the molecular basis of diseases and ultimately aid in the treatment of diseases, which provides an excellent opportunity and foundation for further studies on other models such as cells, mice, and rats to understand the effects of the altered lipid metabolism on human health.

Additionally, owing to the easy-to-handle characteristics and all the available genetic tools and cellular profiles, *C. elegans* are known to be a preferred model for exploring the association between the lipid metabolism and aging. Recent studies have clearly shown that longevity is closely related to the turnover rate and degradation of neutral lipids at specific developmental stages in *C. elegans*. Moreover, Schmeisser et al. described an adipocyte triglyceride lipase (ATGL)-dependent regulation of metabolic homeostasis in muscle tissue, followed by the prolongation of the *C. elegans* lifespan via neuronal AMP-activated protein kinase (AMPK) and a highly conserved nuclear receptor [30]. Nevertheless, there is no direct correlation between the lipid levels and lifespan in nematodes, and simple changes in the lipid levels do not singularly alter their survival. Taking the dietary restriction model as an example, these long-lived nematodes have a lower lipid level due to the blocked feeding, however, the high-fat Insulin/IGF-1 signaling-deficient mutants still have extended lifespans [31,32]. Collectively, the relationship between lipids and a lifespan remains to be explored, and nematodes are the most appropriate and practical model to link these two particularly important physiological indicators.

However, as everything has two sides, several disadvantages of *C. elegans* as the model to study the lipid metabolism cannot be ignored. For instance, there is still no clear classification and definition of lipids. Hence, the total lipid species in nematodes remains a mystery to date, although sterols, phospholipids, and triacylglycerols are particularly common in nematodes. Moreover, compared with other lipidomes, the *C. elegans* lipidome is unique to several lipid classes such as sphingolipids and maradolipids, which makes it different from the mammalian lipid metabolic pathways and brings an inconvenience for the study of the lipid metabolism [33]. In addition, although the nematode genome contains some genes involved in synthesis of cholesterol, it has been thought that *C. elegans* cannot synthesize cholesterol on its own. Therefore, additional cholesterol needs to be added to the nematode growth medium (NGM) for its growth in the experiment.

The pharynx is a neuromuscular pump that acts as the feeding organ responsible for pumping food into the nematode, which is then crushed, ground, and passed to the intestine. The intestine consists of 20 large epithelial cells and is the most important organ for the storage and mobilization of fat. Although nematode has no adipose tissue, it stores fat (triglycerides and cholesterol esters) mainly in lipid droplets (highlighted in a red color) in intestinal and subcutaneous tissue cells.

## 3. Regulation of Fat Synthesis and Degradation in *C. elegans*

### 3.1. Lipid Droplets (LDs)

Due to the absence of dedicated adipose tissues, *C. elegans* store their fat mainly in the intestine [34]. The mobilization of intestinal fat and cellular energy homeostasis are precisely and dynamically regulated in lipid droplets. LDs are delimited by a phospholipid monolayer with neutral lipids such as triacylglycerols (TAGs) and sterol esters (SE) as the core [35]. Neutral lipids are synthesized between the leaflets of the endoplasmic reticulum (ER) membrane via the action of ER-resident enzymes such as stearoyl-CoA desaturases (SCDs) and diacylglycerol acyltransferase (DGAT). The accumulation of synthesized TAGs drives the formation of oil lenses and the latter subsequently grows and buds out towards the cytosol to form nascent LDs [36]. DGAT2 and acyl CoA synthase-22 (ACS-22/FATP1) form a complex at the ER-LD interface to facilitate the expansion of LDs [37]. Except for lipid synthetase, plenty of ER-associated proteins are also crucial for a lipid utilization and LDs biogenesis [38]. For instance, SEIP-1, a homolog of human SEIPIN (a nonenzymatic protein encoded by the Berardinelli-Seip congenital lipodystrophy type 2 gene) [39] can localize to ER-LD contact sites and act as membrane anchors to enable the lipid transfer and LD growth in nematodes [40,41]. COPI (coat protein complex I) components may promote the association of ATGL with the LD surface to mediate lipolysis [42]. Additionally, they have also been reported to transport lipid-producing enzymes from ER to LDs to allow for the subsequent expansion of LDs [43,44].

Importantly, current studies suggest that the size and amount of LDs are proportional to the storage of organismal neutral lipids [17,45]. Accordingly, morphological studies of LDs are beneficial for understanding the storage of abnormal lipids. A recent study has shown that *Stenotrophomonas maltophilia* (*S. maltophilia*) feeding enhanced the DumPY (DPY)-9-dependent ER-LD interaction in the intestine and hypodermis, leading to the formation of supersized LDs in *C. elegans* [45]. In addition, the monomethyl branched-chain fatty acid C17iso could ensure the integrity of the ER membrane, whose biosynthesis requires the acyl coenzyme A synthase encoded by *fatty acid CoA synthetase-1* (*acs-1*), thereby maintaining the function of ER-resident enzymes for an appropriate lipid synthesis and the growth of LDs [46]. However, the growth of LDs was significantly prevented in the *acs-1* RNAi worms due to the reduction in C17iso. Therefore, maintaining ER homeostasis is of a great significance for normal fat levels.

Over the past couple of years, the molecular components involved in the regulation of the size of LDs were successively identified in *C. elegans*. For instance, S-adenosyl methionine synthetase 1 (SAMS-1) linked the synthesis of phosphatidylcholine with the size of the LDs and was crucial for proper lipogenesis, the maintenance of small LDs, and the efficient depletion of a subset of LDs [47]. A saturated forward genetic screen of mutagenized haploid genomes was conducted, from which 118 mutants with supersized intestinal LDs were isolated [48]. Coincidentally, among these genes, *maoc-1* (MAO-C-like dehydratase domain), *dhs-28* (dehydrogenases, short chain), *dauer formation-22* (*daf-22*), and *peroxisome assembly factor-10* (*prx-10*) have been reported to play important roles in the peroxisomal β-oxidation pathway, and their absence may cause the excessive accumulation of TAG [49]. Further results also demonstrated that other mutant genes (e.g., *drop-5* and *drop-9* (lipid droplet abnormal) could alter the size of LDs by affecting their ACS-22-DGAT-2-dependent growth, hydrolysis, or fusion.

In summary, LDs are conserved organelles for the storage and mobilization of neutral lipids, serving as metabolic hubs. The morphological and molecular characteristics of nematodes’ LDs are pivotal to the coordination of lipid homeostasis and the in-depth explorations of the pathways of the lipid metabolism.

### 3.2. Lipogenesis

*C. elegans* mainly obtain fatty acids from the bacteria diet through digestion or an endogenous de novo synthesis in the cytoplasm. As shown in Figure 2, as the rate-limiting step in the synthesis of fatty acids, acetyl-CoA reacts with bicarbonate and ATP to form malonyl-CoA, which is catalyzed by acetyl-CoA carboxylase (ACC/POD-2). By adding two carbon atoms at the carboxyl end stepwise, malonyl-CoA is elongated by fatty acid synthase (FAS/FASN-1) to generate medium- and long-chain fatty acids, mainly palmitic acid (C16:0) [50]. Saturated palmitic acid can be integrated into TAGs or phospholipids, and it can also be modified by fatty acid elongases and desaturases for the biosynthesis of various polyunsaturated fatty acids (PUFAs) [51]. In addition, dietary palmitic and stearic acids can also form PUFA in the same way. Monounsaturated fatty acids (MUFAs) are the key components of membrane phospholipids and TAGs, and the appropriate ratio between them and saturated fatty acids is maintained by the activity of SCDs. As the critical control point in a metabolic regulation, three SCDs promote the synthesis of MUFAs in *C. elegans*, which introduce the first double bond between the 9th and 10th carbon atoms of palmitic acid and stearic acid, converting them to palmitoleic acid (C16:1n7) and oleic acid (C18:1n9), respectively. Among these catalyticase, stearoyl-CoA desaturases exhibit different substrate specificities, FAT-5 shows a preference for palmitic acid, while FAT-6 and -7 prefer stearic acid [52]. Next, linoleic acid (18:2n6) is formed by the Δ12 desaturation of 18:1n9 by FAT-2, while a small amount of 18:2n9 is synthesized by the Δ6 desaturation of 18:1n9 by FAT-3 [51]. After that, another three desaturases (FAT-1, -3, -4) and two elongases (ELO-1, -2) are involved in the endogenous synthesis of C20 PUFAs and other C18 PUFAs [14]. The entire synthesis process ends with the generation of eicosapentaenoic acid (20:5n3).

SCDs are lipogenic enzymes responsible for the generation of MUFAs and energy storage molecules. It is essential for a normal fatty acid composition, the maintenance of the important components of membranes, and the regulation of lipid homeostasis. A phenotypic observation and a fatty acid composition analysis of single mutant strains of *fat-5*, *fat-6*, and *fat-7* showed only subtle changes in the composition of fatty acids without other visible phenotypic changes, whereas the triple mutation of *fat-5*; *fat-6*; and *fat-7* was lethal [53]. This suggests that maintaining a normal fatty acid composition is essential for survival and these three desaturases in nematodes are functionally redundant. Furthermore, the desaturation of fatty acids is an aerobic process mediated by cytochrome b5 reductase, cytochrome b5, and desaturase [54]. HPO-19 and T05H4.4 are cytochrome b5 reductases that affect conversion by the FAT-1, FAT-2, FAT-3, and FAT-4 in *C. elegans* [55]. Additionally, cytochrome b5 CYTB-5.1 is essential for the activity of FAT-6 and FAT-7, whereas CYTB-5.2 is indispensable for the activity of FAT-5 in nematodes [56].

Additionally, monomethyl branched-chain fatty acids (mmBCFAs) are also vital to growth and are synthesized de novo from the branched-chain fatty acids in *C. elegans.* Using isobutyryl-CoA as the substrate, FAS/FASN-1 generates C13:iso, which may be conversed to C15:iso or C17:iso via the actions of the elongases (ELO-5 and/or ELO-6) and 3-ketoacyl-CoA reductase (LET-767).

Surprisingly, lipogenesis is also closely linked to other metabolic pathways in nematodes. The transcription factor PHA-4/FoxA (the defective development of pharynx) acts as a sensor of nucleolar stress to bind to and transactivate the expression of *pod-2*, *fasn-1*, and *dgat-2*, consequently promoting a lipid accumulation [57]. PHA-4 is a mediator that connects the energy metabolism to nucleolar stress tightly. The chemosensory G protein-coupled receptor (GPCR) STR-2 is expressed in AWC and ASI amphid sensory neurons. By regulating the expression of *fat-5*, *-6*, *-7*, and *dgat-2*, STR-2 can control lipid synthesis and ultimately promote the longevity of *C. elegans* [58].

### 3.3. Lipolysis

Lipolysis is a complex and delicate process with multiple enzymatic steps involved, as shown in Figure 2. Consistently, it involves intricate signaling cascades among or within many organelles. Generally, the utilization of stored triacylglycerol and the production of free fatty acids (FFAs) are achieved by a series of lipases. Additionally, the regulatory pathway for the decomposition of FFAs is β-oxidation, an energy-generating process that occurs in the mitochondria and peroxisomes [59]. In *C. elegans*, adipose triglyceride lipase (ATGL/*atgl-1*) and hormone-sensitive lipase (HSL) are the key lipases capable of hydrolyzing acyl esters [60]. Moreover, numerous acyl-CoA synthases (ACS), dehydrogenases (ACD), enoyl-CoA hydratases (ECH), and thiolases play important roles in the β-oxidation pathway.

The glycerolipid/free fatty acid (GL/FFA) cycle links glucose and the lipid metabolism, consisting of lipogenesis and lipolysis segments. The anabolic phase of this cycle is driven by glycolysis-derived glycerol-3-phosphate (Gro3P) and fatty acyl-CoA (FA-CoA) to produce triglycerides (TGs) [61]. Similarly, the lipolysis of TGs is the same as described above, with FFA and glycerol as the end products. As the central metabolite at the crossroads of fat and the carbohydrate metabolism, Gro3P has been found in mammals to be directly converted to glycerol in the presence of glycerol-3-phosphate phosphatase (G3PP) (encoded by *phosphoglycolate phosphatase*, abbreviated as *Pgp*) [62]. Surprisingly, three *Pgp* homologues have recently been identified in *C. elegans* as *pgph* and its protein products has possessed G3PP activity, which is essential for the synthesis of glycerol [63]. The overexpression of *phosphoglycolate phosphatase homolog-2* (*pgph-2*) enhanced the conversion of Gro3P to glycerol, thereby reducing adipogenesis in nematodes. Nonetheless, minimal information related to *pgph-1* was obtained in the experiment, which needs to be explored in future studies.

In mammals, prolonged fasting-induced autophagy is a form of self-protection for energy production. Similarly, many acid lipases regulate the lipid levels in nematodes like mammalian lipophagy. For instance, take, as an example, the overexpression of *lipase like-4* (*lipl-4*) lipase, this results in the production of ω-6 fatty acids during nematode fasting, thereby inducing autophagy, prolonging the lifespan and enhancing starvation survival [64]. Similarly, in *germ line proliferation-1* (*glp-1*) deletion mutants, *lipl-4* lipase is also necessary for the induction of the activation of autophagy [65]. It is thus clear that the lipase action and autophagy are closely related in *C. elegans*, but the in-depth exploration of their relationship is still needed.

Taken together, the processes of the fat metabolism are intrinsically complex, with a large variety of enzymes and their homologs being involved. Therefore, it is necessary to further identify different lipid metabolic pathways to clarify how genes or chemical substances perform their lipid-altering functions.

Nematodes can both obtain fatty acids through the intestinal digestion of dietary lipids and de novo synthesize all its PUFAs from palmitic acid. Then, three molecules of LCFAs and glycerol are esterified to form TGs. The lipids in nematodes mainly exist in the form of TGs, which are decomposed into FAs under the action of lipases. The short, medium, and long-chain FAs are finally converted to acetyl-CoA by a mitochondrial β-oxidation, while the very long-chain fatty acids are finally converted to acetyl-CoA by a peroxisomal β-oxidation. Abbreviations: C13iso, 11-methyldodecanoic acid; C15iso, 13-methyltetradecanoic acid; C17iso, 15-methylhexanoic acid; FAT-1, omega-3 desaturase; FAT-2, Δ12 desaturase; FAT-3, Δ6 desaturase; FAT-4, Δ5 desaturase; FAT-5, FAT-6, FAT-7, Δ9 desaturases; ELO, fatty acid elongase; LET-767, 3-ketoacyl-CoA reductase; TGs, triglycerides; MGAT, monoacyglycerol acyltransferase; DGAT, diacylglycerol acyltransferase; LCFAs, long-chain fatty acids; SCFAs, short-chain fatty acids; MCFAs, medium chain fatty acids; VLCFAs, very long-chain fatty acids; ECH, enoyl CoA hydratase; 3-HACD, 3-hydroxyacyl CoA dehydrogenase; MCAD, medium chain acyl-CoA dehydrogenase.

## 4. Conserved Fat Metabolism Signaling Pathways in *C. elegans*

Research on individual nematode orthologs has revealed that the 12 signaling pathways in *C. elegans* are identical to human signaling pathways, most of which are closely related to the nutrient metabolism [66]. This certainly brings important insights into the function of genes and signaling pathways related to the human metabolism, with important implications for the study and treatment of obesity-related diseases. Accordingly, as shown in Figure 3, several major pathways conserved in humans for controlling the storage of fat in nematodes, such as the insulin-like signaling pathway and the rapamycin signaling pathway, are discussed below with emphasis.

### 4.1. Regulation of Nematode Adiposity Levels by Neuroendocrine Signaling Pathways

To date, evidence from *C. elegans* has shown that the central nervous system plays a profound role in regulating the storage of body fat and energy expenditure, independently of the intake of food. Specifically, environmental fluctuations, sensed and decoded by the nervous system, are the main driving force of the fat metabolism in nematodes [67]. As a common neuromodulator, serotonin (5-hydroxytryptamine, 5-HT) can mediate the neural circuit of fat loss but also regulate the rate and process of the fat metabolism in the uninnervated gut through the tachykinin brain-to-gut signaling axis. In addition, the tachykinin neuroendocrine pathway can integrate other signals and sensing from the nervous system besides 5-HT to achieve the control of the body fat in nematodes.

#### 4.1.1. 5-HTergic Fat Regulation

The remarkable function of the neuroendocrine pathway in controlling lipid homeostasis in response to neural and hormonal signals has been unraveled [68]. As is known to all, the neurotransmitter serotonin, whose functions are conserved in many species, is a major driver of body fat loss and energy expenditure. In mammals, 5-HT can regulate the energy metabolism in multiple cells, such as pancreatic β-cells [69], adipocytes [70], hepatocytes [71], and immune cells [72]. In zebrafish (*Danio rerio*), 5-HT suppresses the weight gain under high-fat diet conditions by activating the lipid catabolism and fatty acid oxidation process primarily [73].

In *C. elegans*, 5-HT is a tryptophan-derived bioamine, whose synthesis is restricted in only a few neurons. Tryptophan hydroxylase TPH-1 is the rate-limiting enzyme in the synthesis of 5-HT, and the loss of TPH-1 may lead to the deficiency of 5-HT and the excessive accumulation of body fat [74,75]. Certainly, 5-HT is also a valuable neuromodulator of the feeding rate and the accumulation of fat. Compared to the control group, 5-HT-treated wild-type nematodes had a significantly lower body fat content but an increased feeding rate [76]. In addition, the decreased production of 5-HT results in increased body fat despite a reduced food intake, whereas an exogenous 5-HT administration promotes a fat consumption despite an increased food intake [77]. Thus, the 5-HT-mediated regulation of fat is independent of the intake of food in *C. elegans*. Serotonergic 7-transmembrane G protein-coupled receptors (GPCR) SER-1, SER-5, and SER-7 have been shown to mediate the effects of 5-HT on the feeding behavior. Specifically, SER-7 regulates pharyngeal pumping and peristalsis, SER-1 is responsible for stabilizing the pumping rate, while SER-5 is an important regulator of the feeding rhythm [78].

However, in the neural circuit for serotonin-mediated fat loss, the 5-HT-gated chloride channel encoded by *defective morphogenesis-1* (*mod-1*) and the GPCR encoded by *serotonin/octopamine receptor-6* (*ser-6*) have been identified as important components of signaling mechanisms, linking neural 5-HT to peripheral β-oxidation pathways [79]. Octopamine (OA) is synthesized in RIC neurons via the tyramine β-hydroxylase encoded by *tyramine beta hydroxylase-1* (*tbh-1*). It stimulates fat loss upon binding to the SER-6 octopaminergic receptor in AWB neurons, thereby modulating the expression of *tph-1* in ADF chemosensory neurons [77]. As the only known ligand, the 5-HT produced in ADF neurons activates MOD-1 in the URX neurons for the further downward transmission of fat-reducing endocrine signals. In the intestine, the nuclear receptor NHR-76 activates the transcription of *atgl-1* after receiving 5-HT-mediated fat loss signals. The fatty acids produced by ATGL-1 hydrolysis then enter the mitochondria and peroxisome for a β-oxidation.

Although the mechanisms of endocrine signaling in neurons and the intestine are well understood, it has remained a challenge to identify the neuroendocrine effectors that relay fat mobilization signals from the nervous system to the intestine, in any system. As the intestine is not directly innervated in nematodes, many secreted neuropeptide ligands bind to corresponding homologous receptors to form the neuroendocrine axis of the 5-HT-dependent fat consumption. The *C. elegans* genome contains a total of 113 neuropeptide genes, which can be grouped to three major families: *FMRF-like peptide* (*flp*), *neuropeptide-like protein* (*nlp*), and *insulin* (*ins*). An RNAi-based suppressor screen for neuropeptide genes and subsequent experiments revealed that *flp-7* is required for the effects of 5-HT on the *atgl-1* mediated fat loss [80]. It is well known that the significant effects of serotonin and food supply signals on nematodes are consistent [81]. Under nutrient-enriched conditions, octopamine signaling and 5-HT signaling are integrated to keep AAK-2/AMPK inactive in ASI neurons, which de-repress the CREB (the cAMP response element binding protein) co-regulator CRTC-1. The increased FLP-7 is subsequently released as dense core vesicles outside the ASI neurons and binds to the neuropeptide receptor 22 (NPR-22) GPCR. Thereafter, endocrine signals induce the hydrolysis of ATGL-1 and fatty acid β-oxidation in the intestine. By using neuroendocrine-related mutants and tissue-specific rescue methods, Eleutheroside E can reduce the intestinal lipid accumulation through serotonin and neuropeptide *flp-7-npr-22* pathways in *C. elegans* [82].

#### 4.1.2. Modulation of Fat by Neuropeptides Unrelated to 5-HT

In addition to the 5-HTergic fat loss, many other neuroendocrine signals have also been identified to regulate the fat levels in the nematodes. However, the potential mechanisms are diverse and complex. On the one hand, many fat-regulatory neurons in nematodes secrete neuropeptides in a manner unrestricted by synaptic wiring so that signals can be transmitted to mobilize or maintain fat. The intestine is the control point where the multiple signals received can be sorted and organized. On the other hand, central regulatory neurons serve as the locus of the control where information from sensory neurons leads to the release of some key neuroendocrine signals that facilitate the synthesis or breakdown of fat.

The discovery in *C. elegans* that an olfactory specificity modulates the lipid metabolism reveals a different neuroendocrine signaling pathway [83]. The olfactory neuron AWC can selectively detect diverse odorant molecules and subsequently relay olfactory information to inter-neurons AIY [84]. The FLP-1 neuropeptides, encased in dense core vesicles (DCV), are then released by AIY and subsequently bind to the NPR-4/neuropeptide receptor. SGK-1/serum- and glucocorticoid-inducible kinase responds to olfactory signaling and activates downstream DAF-16/FOXO to autonomously regulate lipid homeostasis in peripheral fat storage tissues. Together with the decreased fat stores in *flp-1(yn2)* mutants, it seems that *flp-1* is the key to connecting neural signals with the fat metabolism [85].

When *C. elegans* are exposed to low oxygen, neuron BAG releases FLP-17 upon the detection of hypoxic signals, and FLP-17 binds to the EGL-6 G protein-coupled receptor on neuron URX [86]. FLP-17/EGL-6 signaling dampens the resting state of URX neurons, thus inhibiting the oxidation of fat in intestinal cells. Under nutrient-deficient conditions, the FMRFamide neuropeptide FLP-20 is secreted from the AIB neurons in a metabotropic glutamate receptor 2 (MGL-2)-dependent manner and serves as a systemic starvation signal within the nematodes [87]. FLP-20 then acts on the receptor-type guanylate cyclase GCY-28 to promote a lipid degradation in the peripheral tissues.

### 4.2. Insulin/Insulin-like Growth Factor (IGF)-1 Signaling (IIS) Pathway

In *C. elegans*, the insulin-like signal is well-known as a highly conserved neuroendocrine regulator that controls the accumulation of fat to maintain homeostasis. There are 40 insulin-like (INS) peptides in *C. elegans*, which can bind to and subsequently activate only one cell surface transmembrane receptor, DAF-2/INSR [88]. DAF-2 has intrinsic tyrosine kinase activity, whose autophosphorylation activates phosphatidylinositol (PI)3-kinase (AGE-1/AAP-1), leading to the conversion of the membrane PI-bisphosphate (PIP2) to PI-trisphosphate (PIP3) [89]. PIP3 recruits the downstream effector proteins PDK-class protein kinase 1 (PDK-1), AKT kinase 1 (AKT-1), AKT-2, and serum- and glucocorticoid- inducible kinase 1 (SGK-1), which present pleckstrin-homology domains, to the plasma membrane [90]. These activated serine/threonine kinases can phosphorylate and negatively regulate DAF-16/FoxO. In contrast, the lipid phosphatase DAF-18 negatively regulates PI3K signaling. Under such low signaling conditions, active DAF-16 enters the nucleus and transactivates the lipid metabolism-related target genes, such as the Δ9 desaturases genes, fatty acid elongase *elo-2*, and other FAs-synthesis genes, thereby causing the accumulation of lipids [91]. Importantly, insulin signaling and the fat metabolism are mutually regulated. PUFAs, especially eicosapentaenoic acid (EPA), can act as signaling molecules that regulate insulin signaling.

^13^C isotope assays have revealed that the de novo synthesis of fatty acids is the predominant mechanism by which the insulin signaling pathway affects the storage of fat in *C. elegans* [92]. When the environmental conditions are not conducive to normal growth, *daf-2* signaling is downregulated, and nematodes arrest the development at the long-lived dauer diapause stage, storing large amounts of fat instead of transferring energy to the reproductive system [93]. In adult *C. elegans*, the fat content in *daf-2* mutants is significantly increased, and the excess body fat is largely in the form of storage TAGs [75]. Nonetheless, the high-fat phenotype can be suppressed by the inactivation of *daf-16*, which suggests that the increased fat synthesis is dependent on the inappropriate activation of *daf-16*/FoxO. Although both DAF-2 and its mammalian homologs regulate the metabolism, the metabolic defects resulting from their mutations are quite different. Unlike the accumulation of fat in *daf-2* mutants, the complete loss of the mammalian insulin receptor’s activity results in a growth arrest at birth and a metabolic shift to uncontrolled lipolysis and ketoacidosis [94].

The *daf-16*/FoxO genomic locus encodes eight transcripts, and these eight mRNAs are divided into different groups, including *daf-16a*, *daf-16b*, *daf-16e*, *daf-16g*, and *daf-16d/f/h* [95]. The stimulated Raman scattering (SRS) microscopy imaging results showed that the expression of *daf-16a* or *daf-16b* isoform in *daf-2*; *daf-16* double mutants did not restore the lipid levels to those observed in *daf-2* single mutants, whereas expressing *daf-16d/f/h* could restore it [96]. Therefore, the DAF-16D/F/H isoform plays an important role in the fat accumulation induced by the mutation of *daf-2*. Consistent with this finding, the *daf-16d/f/h* isoform was found to autonomously regulate the fat metabolism downstream of olfactory signaling [83]. Moreover, PUFAs also proved to inhibit the nuclear localization and transcriptional activity of DAF-16, suggesting that although *daf-16* was involved in the homeostasis of FAs, its behavior was also controlled by the FAs metabolism [97].

The expression of the majority of insulin-like genes is influenced by the nutrient availability, and the activity of insulin-like peptides such as DAF-28, INS-6, and INS-4 can be clearly up-regulated by feeding [98]. *daf-28* is essential for the regulation of the metabolism after the perception of food. The sensation of environmental polypeptides promotes the transcription and secretion of DAF-28 from ASI and ASJ amphid neurons to activate the IIS pathway and, in turn, affect the lipid metabolism [99]. A recent study has shown that *ins-4* is a negative regulator of the storage of lipids in *C. elegans* by DBL-1/BMP signaling through the IIS pathway. DPP/BMP-like 1 (DBL-1) activates subcutaneous Smad signaling and downregulates the expression of INS-4, which leads to a decrease in the activity of DAF-2/InsR [100]. Diminished insulin signaling activates DAF-16/FoxO in the intestine, which induces a lipid accumulation.

Up to now, in *C. elegans*, many compounds have been identified to regulate the body fat levels through the IIS pathway. An enhanced ROS production can inhibit insulin signaling and then activated *daf-16* stimulates the expression of *fat-5*, ultimately leading to an excessive fat accumulation [101]. The gene expression analysis demonstrated that *Pediococcus acidilactici* CECT9879 (pA1c) could alleviate the metabolic syndrome by reversing the glucose-nuclear-localization of *daf-16*, overexpressing *ins-6*, *daf-2*, or *ageing alteration-1* (*age-1*), and downregulating the key genes for the biosynthesis of FAs [102]. Xiong et al. found that Fuzhuan Tea relied on the regulation of the expression of *fat-7* by DAF-16/FoxO to reduce the adiposity in *C. elegans* on glucose-rich diets [103]. Moreover, as an effective functional component for the prevention of obesity, the extract of *Grifola frondosa* (Maitake) also depended on the significant overexpression of *daf-16*/foxo of the IIS pathway for its fat-reducing activity, and thus could be used for the prevention of the diseases related to the metabolic syndrome [104].

### 4.3. DAF-7/TGF-β-like Signaling

The signaling cascade initiated by the TGF-β-like ligand DAF-7 is a neural sensor of complex environmental conditions coordinating various food-related behaviors and the lipid metabolism in *C. elegans*. The DAF-7/TGF-β ligand secreted only from ASI sensory neurons signals through type I and type II TGF-β receptors DAF-1 and DAF-4 in interneurons RIM and RIC neurons to promote satiation/quietness. Additionally, the signal can also inactivate the receptor-associated co-SMADs DAF-3. However, reduced DAF-7 signaling can activate multiple signaling mechanisms downstream of DAF-3 to regulate feeding, the dauer formation, and fat accumulation. On one hand, activated DAF-3/Smad promotes the production of tyramine (catalyzed by TBH-1) and octopamine (catalyzed by tyrosine decarboxylase TDC-1) from RIM/RIC neurons. Subsequently, these two bioamines signal an inhibition of the pharyngeal pumping rate upon binding to SER-2 receptors on a small subset of pharyngeal neurons, thereby reducing the food intake [105]. On the other hand, the RIM/RIC activation of DAF-3 also conducts the glutamatergic regulation of the accumulation of fat through the MGL-1 and MGL-3 metabotropic receptors located in head neurons [106]. The links between neural MGL-1 and MGL-3 signaling and an intestinal lipid accumulation are still unclear, but the downregulation of DAF-7 signaling initiates a signaling cascade that ultimately promotes a de novo fat synthesis in the periphery [106]. In brief, *daf-7* signaling regulates the fat metabolism and feeding behavior separately through a compact neural circuit in response to complex and variable nutritional conditions.

In general, the quiescence behavior of nematodes is a result of satiety, which leads to the inhibition of the intake of food and movement [107]. It depends on the concentration of food, previous food experience, and the nutritional signals from the intestine [108]. As we all know, delayed satiety, uncontrolled appetite, and the overeating which is associated with this can lead to obesity and metabolic-related diseases [109]. Hence, the proper control of one’s appetite is important to maintain the balance of energy in the body. TGF-β signaling could represent a nematode-favorable food presence feature that mediates quiescence after fasting and refeeding, suggesting that nematodes can properly regulate their feeding according to nutritional status to avoid excessive feeding and the accumulation of fat.

In *C. elegans*, DAF-7 acts as an indicator of the response to environmental conditions. Under adverse environmental conditions, the DAF-7 TGF-β-like ligand is inactivated, the down-regulation of which reduces the pumping rates and feeding rates but leads to an accumulation of intestinal lipids. From the above, it is clear that the regulation of the fat content by *daf-7* is molecularly independent of its modulation of feeding, which is coordinated but an independent output of the nervous system. Therefore, it can be explained that the two results are inversely proportional. The molecular difference eliminates the requirement for the increased feeding rate as the sole basis of an increased fat accumulation [106].

Erythromycin (ERY) is a macrolide antibiotic with obesogenic effects. Recent studies have shown that an ERY treatment significantly down-regulates the expression of *daf-7* and up-regulates the expression of *mgl-1* and *mgl-3*. That is, ERY can not only contribute to overeating by reducing the negative regulation of the TGF-β signaling on feeding but also promote a fat accumulation by stimulating the metabotropic glutamate receptors [110]. In addition, the ERY-induced downregulation of *daf-7* contributes to the decrease in the percentage of satiety quiet and the duration of satiety calm, which can also lead to overeating and thus induce obesity.

PDP1 is a homolog of mammalian pyruvate dehydrogenase phosphatase (PDP) in *C. elegans* and acts as a component of the DAF-7/TGF-β pathway. Including fat storage, PDP1 can regulate multiple phenotypes of the IIS pathway by positively regulating DAF-16. Under poor survival conditions, such as food restriction, PDP-1 negatively regulates TGF-β signaling and then activates Co-SMAD DAF-3 and SNO-SKI repressor DAF-5 to inhibit the transcription of agonistic insulins or promote the transcription of antagonistic insulins. As a result, signaling through the IIS pathway is reduced and the nuclear localization of DAF-16 is enhanced, which in turn leads to the increase in the fat storage [111]. Thus, the TGF-β and IIS pathways can link together through PDP-1 and jointly regulate the lipid metabolism in *C. elegans*.

### 4.4. TOR

The target of rapamycin (TOR) is an evolutionarily conserved protein with a serine/threonine kinase activity that regulates cellular energetics, growth, and the metabolism. TOR functions as the catalytic core of two structurally and functionally distinct multiprotein complexes, TOR complex 1 (TORC1) and TOR complex 2 (TORC2). The proteins Raptor and Rictor are mutually exclusive binding partners for TOR and define the TORC1 and TORC2 complexes, respectively [112]. In *C. elegans*, RICT-1 is the homolog of Rictor, which has been shown to regulate the energy metabolism and lipid accumulation in TORC2. Mutations in *rict-1* result in an inappropriate high-fat phenotype. As the targets of TORC2 phosphorylation, Akt kinases are considered to be key mediators of TORC2 signaling. However, whether RICT-1/TORC2 depends on Akt signaling to regulate the storage of fat is still unsettled issues [52]. Instead, the genetic analysis of the mutants has shown that *rict-1* mutants share many phenotypes with AGC kinase *sgk-1*, the sole *C. elegans* homolog of the serum and glucocorticoid-induced kinase gene family, and the overexpression of wild-type SGK-1 can suppress the fat mass in *rict-1* mutants [113]. These results indicate that SGK-1 is the key downstream kinase and the critical physiologically significant mediator of the lipid storage effect of TORC2. In addition, it is known that DAF-16 is the only genetically and biochemically defined target of the AKT and SGK kinases in nematodes [114], but the phenotypes caused by the inactivation of *CeRictor* are not mediated by DAF-16, thus TORC2 regulates the storage of fat completely independently of DAF-16.

The dosage compensation complex (DCC) plays an important role in balancing the expression of X-linked genes between males and females [115]. Recently, the knockdown of ten DCC members could reduce the elevated fat mass to normal levels in *rict-1* mutants, suggesting that the DCC acted downstream of TORC2/rict-1 to negatively regulate the metabolism [116]. The DCC component DPY-21 physically associates with SGK-1 proteins, demonstrating that TORC2 may directly regulate the DCC through the phosphorylation of DPY-21. Therefore, DCC is the novel downstream negative regulator of the TORC2/SGK-1 pathway, which links the metabolism to the chromatin status in *C. elegans*. mTORC2 and SGK-1 also mediates the lipid metabolism in the intestine by controlling the activity of the PQM-1 transcription factor coupled to the hypodermal microRNA (miRNA) pathways, thereby facilitating the inter tissue transport of the fat reserves from somatic cells to the germline [117].

In addition to modulating the production of lipids directly in the intestine, TORC2 also conveys the metabolic state information and internal nutrition signals to the brain, thereby altering hormonal signaling by the nervous system in *C. elegans*. The TORC2-dependent pathways that transmit nutrient information directly or indirectly between the gut and head sensory neurons of nematodes are currently unknown [118]. However, intestinal TORC2 signaling targets *daf-7* and *daf-28* in sensory neurons to regulate the dauer formation and may also target other neurons to modulate food-dependent foraging decisions in adult nematodes. TORC2 functions as cell-intrinsic sensors and regulators of the metabolic status to allow worms to regulate behavioral responses, such as feeding, in response to different and variable environmental conditions based on the metabolic status.

### 4.5. AMPK

AMPK, an evolutionarily conserved heterotrimeric serine/threonine protein kinase complex with one catalytic (α) and two regulatory subunits (β and γ) [20], acts as a cellular energy sensor in eukaryotes. AMPK can be activated by the increased ratio of AMP to ATP or nutritional deficiency signals. Additionally, it restores the energy balance and lipid homeostasis by negatively regulating energy-using anabolic processes such as lipogenesis and positively regulating energy-producing catabolic processes., e.g., fatty acid β-oxidation and glycolysis [119]. The regulation of the energy balance and body fat stores by the AMPK is conserved from yeasts to humans. In mice, stevioside can ameliorate antiadipogenic effects and promote a β-oxidation in adipocytes by activating AMPK-mediated signaling [120]. In zebrafish, the PH domain and leucine-rich repeat protein phosphatase 1 (PHLPP1) promotes the accumulation of neutral lipids through the AMPK/ChREBP-dependent lipid uptake and fatty acid synthesis pathways [121].

In *C. elegans*, AMPK functions to downregulate lipolysis in the hypodermis during dauer to ensure the slow release of stored energy that enables it to be maintained until diapause [122]. Two genes, *AMP-activated kinase-1* (*aak-1*) and *aak-2,* encode different catalytic α-subunits of AMPK. Among them, *aak-2* functions with either of the two β regulatory subunits to maintain the lipid stores and reserve energy during dauer. This is why dauer larvae die from the rapid and premature consumption of energy stores after the loss of the *aak-2* function [122]. Although AMPK directly phosphorylates hormone sensitive lipase (HSL) in rats, ATGL-1 is the direct phosphorylation target of AMPK at multiple residues in *C. elegans*. During the dauer stage, the phosphorylation of AMPK leads to the generation of 14-3-3 binding sites on ATGL-1, which are recognized by the 14-3-3 protein orthologue PAR-5 in *C. elegans* [123]. The physical interaction of ATGL-1 with PAR-5 blocks ATGL-1 from approaching the triglyceride substrate encapsulated in the lipid droplets, limiting lipolysis and ultimately protecting the cellular energy stores from a rapid depletion.

Treatments of sublethal stresses such as starvation early in life can prolong the nematode lifespan, but mutations in *aak-2* greatly reduce or even eliminate the lifespan extending effects of these stresses. Therefore, AAK-2 appears to be required for the perception of starvation and other stresses early in life, which also leads to a longer lifespan [124]. Using a genome-wide RNAi screen for fat regulatory genes in *C. elegans*, Webster et al. defined a proteostasis-AMPK signaling axis that is central to organismal starvation defenses [125]. AMPK is a key determinant of starvation survival influenced by proteasome homeostasis. Via RNAi, the knockdown of cytoplasmic aminoacyl tRNA synthetases (ARS genes) results in an enhanced proteasomal activity in nematodes, which would activate AMPK in response to low cellular energy stores. Subsequently, the activation of AMPK further resulted in an increased lipid synthesis, decreased lipid oxidation, and extended starvation survival.

AMPK is a recipient of metabolic signals in peripheral tissues but also functions as an upstream modulator of hormonal pathways by regulating their neural secretions. Firstly, AMPK can link serotonergic signaling to the release of glutamate to regulate the feeding behavior in *C. elegans* [81]. The serotonin signaling initiated from ADF neurons acts on the SER-5 serotonergic receptor on the AVJ pair of interneurons, which may activate protein kinase A (PKA). Subsequently, PKA inactivates AAK-2 by inhibiting phosphorylation on AAK-2 (Serine 244). AMPK also affects the glutamatergic release, which regulates the pharyngeal pump rate. On the other hand, high levels of 5-HT promote the secretion of systemic regulators of pro-growth and differentiation pathways by inactivating AAK-2 [126]. The loss of the neural AMPK exhibits similar effects on the fat and hormonal secretions as elevated serotonin. Therefore, several physiological phenotypes once thought to be caused by the activity of AAK-2 in peripheral tissues may be the role of AMPK in neural 5-HT signaling.

### 4.6. Transcriptional Regulation of Lipid Metabolism in C. elegans

In addition to the above pathways, the adiposity levels in *C. elegans* are regulated by many transcription factors and corresponding transcriptional regulators. As ligand-activated transcription factors, nuclear receptors play important roles in regulating the lipid metabolism, energy production, and homeostasis by establishing a link between signaling molecules and transcriptional responses, making them ideal targets for the treatment of obesity-related diseases. In *C. elegans*, nuclear receptors can also synergize with co-regulators to regulate the function of the sterol regulatory element binding protein (SREBP), a body fat transcriptional regulator conserved in both nematodes and mammals. Some of the major components of the transcriptional regulation of the nematodes’ lipid metabolism will be described in detail below.

#### 4.6.1. NHR

Some members of the nuclear hormone receptor (NHR) family of transcriptional regulators function as sensors of the intracellular fatty acid levels and regulators of the lipid metabolism in *C. elegans* and mammals. The mammalian genome encodes a total of 48 NHRs, but only several of which can interact with fatty acids and other lipids to regulate the energy balance, including the peroxisome proliferator-activated receptors (PPARs), liver X receptor, and farnesoid X receptor [127]. In particular, PPARα is a central regulator of energy homeostasis in metabolic tissues, whose activation markedly promotes the uptake, utilization, and breakdown of FAs, leading to the decreased serum levels of TAGs and FFAs [128]. Although *C. elegans* contains 284 NHRs, only 15 of which are conserved in mammals. Of note, none of the 15 genes are associated with the fat metabolism and of the remaining 269 NHRs, NHR-49 is functionally close to PPARα, although it shares a strong sequence similarity with the mammalian hepatocyte nuclear factor 4 (HNF4) receptors.

In fact, *nhr-49* serves as a key regulator of fat usage, modulating two different metabolic pathways that control the degradation of stored fat and maintain the homeostasis of the saturation of fatty acids [127]. Specifically, *nhr-49* stimulates the consumption of fat mainly by promoting the expression of fatty acid β-oxidation-related genes, such as acyl-CoA synthetase gene *acs-2*, mitochondrial β-oxidation trifunctional enzyme gene *ech-1*, and carnitine palmitoyl transferase gene *F09F3.9*. The quantitative RT-PCR screening revealed that the deletion of *nhr-49* displayed a greater effect on the mitochondrial β-oxidation than peroxisomal β-oxidation. Therefore, *nhr-49* has multiple target genes involved in the transport of fatty acids into the mitochondrial matrix. On the other hand, *nhr-49* modulates the fatty acid composition by promoting the expression of SCDs, mainly on stearic and OA levels, but not on the abundance of PUFAs. However, *fat-7* is involved in a negative feedback mechanism that represses the expression of β-oxidation genes, independently of NHR-49, possibly by signaling through an as-yet-unidentified fatty acid species. That is, by directly stimulating *fat-7*, *nhr-49* indirectly inhibits the same set of genes that it independently activates, acting to hinder the fat degradation. Moreover, based on the function of FAT-7, C18:0 can act as an activator of the β-oxidation gene’s expression, while C18:1n9 can act as a blocker or both. *Nhr-49* selectively regulates the two pathways that consume fat for energy and partition fat for storage, in response to the changing energy demands due to complex environmental conditions.

By simultaneously promoting two distinct branches of the FA metabolism pathways, mitochondrial β-oxidation and FA-Δ9-desaturation, NHR-49 helps to accommodate the loss of the reproductive potential and establish a lipid profile that facilitates a lifespan extension. It was shown that NHR-49 promotes the longevity of germline-less adults by eliminating the lipids designated for reproduction and converting saturated fatty acids (SFAs) into unsaturated fatty acids (UFAs) that are more conducive to cellular maintenance [129]. In addition, NHR-49 modulates the immunometabolic axis of survival in *C. elegans* by controlling the lipid catabolism as well as inhibiting the key enzymes involved in the synthesis of neutral lipids and the production of immune effectors upon an exposure to *Enterococcus faecalis* (*E.faecalis*) [130,131].

By binding to long-chain fatty acids, fatty acid-binding proteins (FABPs) facilitate the cellular uptake and transport of fatty acids in mammalian tissues and target them to specific metabolic regulatory pathways. As one conserved homolog of FABPs in *C. elegans*, LBP-5 can regulate the energy metabolic function of NHR-49 and directly bind to various fatty acids with different affinities. Usually, LBP-5 and NHR-49 work together as functional partners in the lipid metabolism. LBP-5 can affect the expression of the related genes in an *nhr-49*-dependent manner to promote a fatty acid desaturation/elongation, mitochondrial/peroxisomal β-oxidation, and enhance the fatty acid gluconeogenesis [19]. In addition, NHR-49 may enhance the expression of *lbp-5* to promote the transport of fatty acids. Additionally, LBP-5 reduces glycolysis in the life cycle of a normal nematode. In *lbp-5* mutant worms, glycolysis is activated to compensate for the energy shortage due to the insufficient mitochondrial β-oxidation of FAs [132].

Another *C. elegans* transcription factor NHR-80 has also been identified as a regulator of the SCDs expression, which is crucial for maintaining the normal unsaturation of FAs and a proper membrane composition. In addition to activating the expression of the SCDs genes, NHR-80 can increase the expression of other desaturases in the absence of FAT-6 or FAT-7. Although NHR-80 regulates the composition of fatty acids, it does not alter the level of fat storage or the expression of β-oxidation-related genes. NHR-49 and NHR-80 are derived from the same ancestral gene, and they are both required for the SCDs expression. Nevertheless, NHR-80 does not alter the level of fat storage or the expression of β-oxidation-related genes when regulating the composition of fatty acids. By comparison, NHR-49 regulates lipid homeostasis pathways more broadly.

#### 4.6.2. LPD

In mammals, sterol regulatory element binding proteins (SREBPs) are ER membrane-embedded transcription factors and are vital for the uptake, biosynthesis, and oxidative catabolism of lipids. The SREBP family consists of three proteins, SREBP-1a, SREBP-1c, and SREBP-2. Among these proteins, SREBP-1 preferentially activates the genes of the fatty acid metabolism, while SREBP-2 preferentially activates the genes of the cholesterol metabolism [133]. Although both SREBP-1a and SREBP-1c regulate the genes involved in the synthesis of MUFAs and PUFAs and their incorporation into TAGs and phospholipids, SREBP-1a can be involved in the transcriptional regulation of cholesterol synthesis [134]. *C. elegans* possess a single SREBP (SREBP-1c) ortholog, Y47D38.7/sbp-1/lpd-1, which is expressed in all metabolic tissues, especially in the endodermal fat-storing cells and intestine. Similar to the function of SREBP in mammals, SBP-1/LPD-1 is also required for a lipid accumulation in *C. elegans*. In *sbp-1* knockdown worms, the fat storage and body size were significantly reduced, similar to the characteristics of starvation and the presence of *eat-2* mutants [135]. SBP-1 is a key modulator of fatty acid synthesis and desaturation, contributing to the maintenance of lipid homeostasis in *C. elegans*. The deletion or mutation of *sbp-1* results in the downregulated expression of lipid synthesis genes and the upregulated expression of lipid degradation genes. Among the seven genes encoding fatty acid desaturases, *fat-2*, *fat-5*, *fat-6*, and *fat-7* are regulated by SBP-1, whereas *fat-1*, *fat-3*, and *fat-4* are either controlled by other factors or under the joint regulation of multiple mechanisms. In addition to straight-chain fatty acids, SBP-1 also controls the synthesis of mmBCFA by regulating the long-chain fatty acid elongases *elo-5* and *elo-6*, as well as the very long-chain acyl-CoA synthase *acs-1*.

SBP-1/LPD-1 (lipid depleted-1) is named as the nematode phenotype of lacking fat in the absence of the SREBP homolog. For the same reason, *cebp-2/lpd-2* was chosen as the genetic name for the homolog of mammalian CCAAT/enhancer-binding proteins (C/EBPs) in *C. elegans* [136]. C/EBPs are necessary for an adipocyte differentiation and maturation; they are major players in the modulation of lipogenesis and adipogenesis as well as gluconeogenesis [137]. CEBP-2 is a crucial regulator of the normal lipid metabolism in *C. elegans*. The inhibition of RNAi or the mutants corresponding to this gene have been shown to produce a low lipid accumulation, which is attributed to abnormalities in the fatty acid mitochondrial β-oxidation and fatty acid desaturation [138]. *cebp-2* affects the fat storage mainly by regulating the expression of genes for a fat degradation, such as *ech-1.1* and lipogenic enzyme genes such as *fat-5* in *C. elegans*. As mentioned above, ECH-1.1 accelerates the consumption of fat for the production of energy during fatty acid β-oxidation, while FAT-5 promotes the production of palmitoleic acid. However, it is uncertain whether the two genes function independently or are related to each other.

#### 4.6.3. MDT-15

MDT-15/MED15, a mediator complex subunit, is an interacting protein and transcriptional co-regulator of the transcription factors NHR-49 and SBP-1. MDT-15 regulates the lipid metabolism in response to changes in the metabolic status and is therefore considered as an important node in the regulatory network that maintains metabolic homeostasis. MDT-15/MED15 confers the regulation of the fatty acid metabolism through direct and specific binding to the activation domains of NHR-49 and SBP-1. In well-fed nematodes, MDT-15 synergizes with SBP-1 to promote the fat storage and adipogenesis [139], while in fasted nematodes, MDT-15 would be selected to effectively respond to short-term fasting together with NHR-49 [140]. In addition to monitoring the energy supply and regulating the metabolism, MDT-15 is also required for the regulation of the response to environmental toxicants or various stresses. In *C. elegans*, the signaling cascade of MDT-15-SBP-1 mediates a modest increase in the fat storage by activating FAT-6 in the intestine, thus forming a protective response to the toxicity of the simulated microgravity [141]. In addition, MDT-15 can also modulate the expression of detoxification genes that respond to environmental toxicants to monitor the quality of ingested substances and facilitate the elimination of harmful exogenous substances [142]. These transcriptional regulatory pathways integrated by MDT-15 are coordinated and interrelated and they contribute to the screening, utilization, and elimination of ingested substances by nematodes.

Furthermore, lipid homeostasis regulated by MDT-15 is necessary for longevity. MDT-15/MED15 can upregulate the expression of fatty acid desaturases and facilitate the conversion of SFA to UFA. The maintenance of a high UFA/SFA ratio in *C. elegans* prevents worms on glucose-enriched diets from accelerated aging [143,144], which is also critical for low-temperature-induced longevity and proteostasis [145].

### 4.7. PRY-1/Axin Signaling

The mammal Axin homolog was initially identified as a negative regulator of the WNT-mediated signaling pathway [146]. As a member of the Axin family, PRY-1 acts on regulating the fat metabolism and tissue homeostasis in *C. elegans*. Oil red O staining showed that the *pry-1* mutants had a decreased fat storage and an altered lipid distribution [147]. The transcriptome profiling of the *pry-1* mutants revealed that multiple lipogenic genes (*fat-5* and *fat-6*), four lipase genes (*lips-3*, *lips-7*, *lips-10*, and *lips-17*), as well as two lipid-binding protein genes (*lbp-5* and *lbp-8*) were downregulated completely [147]. Apparently, *pry-1*-mediated signaling is mainly involved in the lipid metabolism and energy utilization. Furthermore, RNAi and the transcription level analysis indicated that *pry-1* may act in parallel with *nhr-49* and *nhr-80*, but the possibility of interacting with *sbp-1* to affect the expression of the downstream adipose gene could not be ruled out. Of note, the knockdown of *vit* genes, which encoded vitellogenins, could inhibit the low lipid phenotype of the *pry-1* mutant. More importantly, *vit-2* was the only gene of the *vit* gene family which was significantly upregulated in *pry-1* mutants, suggesting that *vit-2* was negatively regulated by *pry-1* [148]. In addition to the fact that exogenous OA could restore the lipid content of the *pry-1* mutant to a normal level, the gas chromatography (GC)-MS analysis also indicates that *pry-1* is required to maintain the levels of the beneficial fatty acid species analyzed [147]. However, whether *pry-1* is involved in other mechanisms to exert the lipid regulation in *C. elegans* as well as its other interacting genes and downstream effectors in this pathway still needs to be further investigated and discovered.

## 5. Experimental Tools for the Studies of Fat Metabolism in *C. elegans*

Lipids are important substances for the storage and supply of energy in *C. elegans*, and their metabolism is an important and complex biochemical reaction in the body. Therefore, the characterization and study of the nematode lipid levels using intuitive and appropriate experimental tools and methods can help monitor the nematode physiological functions and are of a great importance for their life activities. In the past decade, the lipid assay techniques of nematodes have developed rapidly and made great contributions to metabolic studies. As shown in Figure 4, this review will then describe in detail the experimental tools for the nematodes’ lipid metabolism studies in terms of the label-free visual imaging, biochemical and chemical assays, and staining analysis of lipids.

This figure shows the specific techniques for the nematode lipid analysis, which can be performed by referring to the listed literature. The study of lipids is not limited to a single experimental method but can be best achieved through the permutation and combination of the multiple techniques shown in the figure.

### 5.1. Staining-Based Methods to Characterize Lipid Storage in C. elegans

The transparent nature of the nematode body allows for the characterization of its fat distribution by staining. Using specific staining reagents to stain the adipose tissue of *C. elegans* enables an accurate localization, observation, and relative quantification of the lipids. The fixation-based Oil Red O, Sudan Black B, and the fat-soluble dyes Nile Red and BODIPY used for feeding live nematodes have all been shown to be good fat stains, which are widely used in lipid metabolism studies.

#### 5.1.1. Vital Dye (Non-Fixative) Staining Analysis

Nile red (NR), 9-diethylamino-5H-benzo[α]phenoxazine-5-one, is an excellent stain for the detection of an accumulation of lipids and is capable of visualizing lipid storage droplets in living nematodes because it is an intensely colored, uncharged heterocyclic molecule that dissolves well in lipids and has a negligible interaction with the surrounding tissues. It is often used as a fluorescent probe to label the lipid storage in LDs or lysosomes, monitor cholesterol and lipid order selectively in biomembranes, and distinguish the neutral lipids from phospholipids or other amphiphilic lipids [149,150,151]. The maximum excitation and emission wavelengths of NR are 450–500 and 520 nm, respectively [152]. The addition of NR to nematodes’ food *E. coli* OP-50 leads to the uptake and incorporation of the dye into LDs in intestinal cells without affecting their growth rate, brood size, feeding, or lifespan [20]. Since this method is conducive to the measurement of the fat level and live imaging of *C. elegans*, plenty of the genes and compounds involved in the fat metabolism and its regulation have been screened and identified [75,153,154,155]. Although this method is currently widely used, it is very sensitive to many genetic and exogenous environmental manipulations that can affect the fat content. Recently, Raman-activated Ag nanoparticles combined with NR for in vivo imaging were developed as sensitive and highly specific lipid-targeting Raman probes [156]. Such probes can be incorporated into intracellular intestinal granules by the uptake of nematodes, which in turn visualizes LDs and detects a Raman signal via a conventional confocal microscope. In recent years, NR staining after a brief fixation has been used more frequently for the rapid and accurate detection of neutral LDs in *C. elegans* [157]. Post-fixation staining enables the more complete penetration of the worms into the NR staining solution, which is not dependent on the ability of nematodes to uptake or concentrate the dye [158]. Using this method, Pentagalloyl Glucose [159], the Recombinant buckwheat trypsin inhibitor [160], *Borago officinalis* seed oil [161], and many other compounds have been shown to significantly alter the fat storage in *C. elegans*.

BODIPY (4,4-difluoro-4-bora-3a,4a-diaza-s-indacene) is an intrinsically lipophilic fluorophore with lower background staining and narrower emission spectra. Hence, BODIPY is often used as a vital marker for the qualitative and quantitative measurements of fat in *C. elegans*. Unlike NR, BODIPY is conjugated to a fatty acid moiety, which makes it more specific than NR for staining LDs [162]. In *C. elegans*, after mixing with *E. coli* bacteria, BODIPY 493/503-based fat staining and flow cytometry approaches enable the labeling of LDs in vital worms. This method preserves the worm’s anatomy and the native morphology of LDs but also allows for the quantification of the fat content per volume at the individual-worm level, facilitating the accurate screenings of fat storage mutants [163]. Several studies have used BODIPY-labeled FAs in nematode models to probe the fatty acid uptake and identify regulators of the fat mass [164,165,166,167]. Recently, many efficient probes labeling lipids such as benzothieno [7,6-b]-fused BODIPYs have been designed and synthesized for live cell imaging [168,169,170]. However, the application of these BODIPY-based probes in the *C. elegans* lipid detection remains to be further investigated.

Notably, although simple and inexpensive to perform, many contradictory findings suggest that NR or BODIPY fatty acids cannot serve as accurate proxies for the major fat stores in nematodes. Taking the fat content detection of *daf-2* mutants as an example, the decreased NR or BODIPY-labeled fatty acid staining phenotype is completely inconsistent with the increased fat storage revealed by a biochemical lipid quantification [171]. Therefore, it is best to use the appropriate fat assessment methods according to the actual situation to identify the genes and signals that regulate the energy balance in nematodes.

#### 5.1.2. Fixative-Based Assays

Oil Red O (ORO), formerly known as Sudan Red 5B, is a fat-soluble diazol dye that specifically stains and detects neutral lipids and cholesteryl esters but not biological membranes [172]. ORO has been frequently used for the qualitative and quantitative analysis of mammalian adipocytes [173,174]. The recurring correlation between lipid biochemistry and ORO proved that ORO is a reliable proxy for the fat mass in *C. elegans* [171]. ORO-stained LDs stand out against the transparent body of the nematodes, which helps to qualitatively assess the distribution of lipids between different tissues such as the hypodermis, intestine, and germline. ORO is also capable of staining the neutral lipid deposits in frozen sections, and the color image segmentation of ORO-stained and hematoxylin counter-stained sections produce reproducible proportional data that reflect the overall lipid content in individual nematode [175]. By ORO staining, many of the transcription factors and compounds capable of altering the nematode fat storage and metabolic homeostasis have been identified and studied [176,177,178]. Furthermore, the combination of ORO with NR can quantify the total lipid levels, serving to determine the alteration of the accumulation of lipids in nematodes when changes in the genetic background or feeding conditions occur. To better assess the genes affecting the body fat mass and distribution on a genomic scale, a radically improved whole-animal fat scoring protocol, quick Oil red O (qORO), was developed based on the intrinsic ORO staining method [179]. This method of processing 96-well plates of RNAi or compound-treated nematodes in only 15 min of hands-on time enables an efficient and rapid fat assessment. Nevertheless, as a fixative-based non-fluorescent dye, ORO requires fixed animals and laborious staining procedures but may also fail to distinguish neutral lipids from autofluorescent granules such as lipofuscin.

Sudan Black B ((2,2-dimethyl-1,3-dihydroperimidin-6-yl)-(4-phenyldiazenylnaphthalen-1-yl)diazene) is a fat-soluble diazo dye, which is widely used to visualize various lipids such as sterols, phospholipids, and neutral triglycerides. However, unlike other Sudan dyes, Sudan Black B is not lipid-specific and can also be used to stain other cellular components such as chromosomes, leukocyte granules, and Golgi bodies [180,181,182,183]. Sudan Black B can also indicate lipid levels by staining lipids with an opaque blue-black color in *C. elegans*. After treating the starved worms with a paraformamide fixation for several hours, they are sequentially dehydrated by washes in ethanol of a low to high concentration, and finally stained and de-stained. This simple and effective method can be used instead of coherent anti-Stokes Raman scattering (CARS) microscopy to reflect the whole fat content in the organism [157]. Using Black B, the inhibitory effects of taurine [184] and hesperidin [185] on the accumulation of fat and the adiposity-inducing effect of the transcription factor HIF-1 (hypoxia inducible factor 1) under hypoxic conditions [186] were identified. However, when performing the final de-stained process with ethanol, the operating time greatly affects the results. In addition, slight variations can greatly interfere with the final intensity of the Sudan Black signal, ultimately leading to inter-sample variability [171].

### 5.2. Label-Free Visual Imaging

The above lipid labeling methods based on staining and fluorescence imaging are often interfered by the intrinsic properties and normal function of biomolecules and may suffer from the physical and chemical environment in *C. elegans*. Consequently, convenient, non-invasive, and label-free methods are extremely necessary to achieve rapid imaging and the accurate quantification of the storage of lipids [187].

#### 5.2.1. Non-Linear Imaging Techniques

Non-linear microscopy techniques that exploit the intrinsic properties of the sample for an image contrast have been developed and widely used. Among them, the CARS microscopy technique is capable of the chemically selective and highly sensitive imaging of lipids and provides molecular-specific insights into the lipid metabolism and storage of lipids, and thus has been successfully employed to record the *C. elegans* lipid profile [188]. In CARS microscopy, two laser beams with different frequencies, pump/probe and Stokes, focus on a common focal spot on the specimen [189]. When the beating frequency is tuned to c.a. 2845 cm^−1^, the energy difference in the lasers matches the symmetric stretch vibration of the CH_2_ groups, and lipid-rich structures that are abundant in CH_2_ groups produce a strong CARS signal [190]. Thus, the size and 3D distributions of the lipid accumulation can be monitored and visualized without labeling [188]. In the bythe high resolution of CARS microscopy, many chemical substances such as copper [191], proanthocyanidin trimer gallate [192], macrocyclic lactone ivermectin [193], and polystyrene microbeads [194] have been revealed to modify the distribution of the desaturation or storage of lipids in *C. elegans*. Additionally, CARS imaging can also be used in conjunction with other techniques to visualize and quantify the fat stores of nematodes preferably. The combination of CARS imaging with spontaneous Raman microspectroscopy enables the evaluation of the unsaturation ratio in the single LDs in live worms [195]. In another study, the effects of PUFAs on the homeostasis of the yolk lipoprotein in *C. elegans* were investigated using CARS and two-photon excitation fluorescence (TPE-F) microscopy [196]. CARS microscopy is more sensitive for the detection of the abnormal accumulation of nematode yolk lipoproteins compared to the NR staining, which is similar to the finding that NR-stained structures only partly co-localized with the CARS signals from LDs [197]. Hence, CARS that is applicable to live worms without labeling eliminates the invasive steps such as fixation and staining, but also allows for the more sensitive detection of the accumulation of fat in nematodes. There is no doubt that the CARS microscopy technique provides easy-to-operate, high-speed, and sensitive ways to study the lipid metabolism in *C. elegans*. However, it is costly due to the need for advanced equipment. Strong autofluorescence and non-resonant signals arising from any objects, or the surrounding solvent, can limit the image contrast as well as the spectral sensitivity of the CARS measurements. Moreover, there is a complex non-linear relationship between the CARS signals and analyte concentration [198].

Compared to CARS microscopy, SRS microscopy is more widely used for lipid imaging because it is free from both autofluorescence and non-resonant backgrounds and offers a strict linear dependence on the molecular concentration [96]. Using SRS microscopy, the CH_2_ bonds of the lipid molecule absorb some of the energy of the incident photon (known as the “pump” photon) and are excited to a higher vibrational energy level, resulting in the boosted rate of the scattered photon (known as the “Stokes” photon) generation when measuring LDs [199]. As a result, two signals demonstrate that an increase in the intensity of the “Stokes” beam (Stimulated Raman Gain, SRG) and a loss in the intensity of the “pump” beam (Stimulated Raman Loss, SRL) are generated, which can in principle be used for the SRS microscopy imaging of the lipid molecules. SRS microscopy can distinguish different classes of lipid molecules based on their characteristic Raman shifts and is capable of mapping the three-dimensional (3D) distribution of chemical bonds within the thick tissue samples, benefited from its two-photon optical processes [189,199]. Based on the ability to identify, quantify, and detect the distribution of lipids, SRS microscopy has been applied in *C. elegans* to high-throughput screens or in combination with an isotope tracing assay to quantitatively track the lipids’ synthesis and mobilization [200,201]. Using label-free SRS microscopy, Wang et al. performed RNAi screening based on the quantitative imaging of lipids and identified multiple genetic regulators of the regulation of fat under physiological conditions [202]. Novel roles for BMP signaling in linking mitochondrial homeostasis and the lipid metabolism were also discovered using photo-highlighting on SRS microscopy for the identification of mutants with an altered lipid distribution [203]. Although SRS microscopy is label-free and sensitive, its detection system is subject to a cross-phase modulation (XPM), caused by the strong interaction between the laser pulse and the samples [204]. The limited penetration depth in highly scattering tissues, requirements for professional expertise and operation, substantial costs, and technical complexity of SRS microscopy also limits its extensive application in laboratories.

In addition to the relatively widely used CARS and SARS microscopes, some other nonlinear microscopy techniques can also be used to visualize and quantify the lipids in *C. elegans*. Third harmonic generation (THG) has been reported for the visualization of the LDs in *C. elegans* [205]. The THG image demonstrated the distribution of lipids in the worms, which was supported by multiple CARS signals [206]. Together with second harmonic generation (SHG), THG microscopy has been used to record 3D imaging and monitor the ectopic accumulation of lipids in the non-adipose tissues of nematodes, such as muscular areas [187]. Requiring only one pulsed laser light source, THG is a convenient and cost-effective imaging method for the deposition of nematode lipids, but its sensitivity for the detection of lipids in different biological samples remains to be determined [207]. In *C. elegans*, two-photon excitation fluorescence (TPE-F) microscopy can also be used to characterize lipids. Mari et al. initially used TPE-F to localize the NR-stained lipids [187]. Coupling THG and TPE-F modes in a single microscope also enables the simultaneous visualization of non-fluorescent neutral lipids and autofluorescent aggregates [207]. Furthermore, TPE-F microscopy can be used together with CARS microscopy to examine the transport of yolk lipoprotein [196].

#### 5.2.2. LD-Associated GFP Reporter-Based Assay

Due to the transparent nature, GFP-tagged LDs-specific markers have been widely used in the lipid studies of *C. elegans*. By detecting the fluorescence of GFP, the storage of lipids and the expression patterns of the genes related to the lipid metabolism can be visualized [208,209,210,211].

The perilipin (PLIN) family proteins are widely used as LD-specific markers in mammals and *Drosophila* for their ability to specifically localize to LDs [212]. Although the apparent homologs of PLIN proteins in nematodes have not been identified, PLIN proteins can localize to LDs across species boundaries [213,214]. Thus, by expressing *Drosophila* PLIN1::GFP in *C. elegans*, PLIN1::GFP coated LDs and could be used as a fat storage indicator in living worms [215]. The abundance and size of PLIN1::GFP-labeled LDs were closely correlated with the fat storage status, which was successfully validated by NR staining. By the PLIN1::GFP marker, it was possible to record the LD dynamics in the whole live nematodes under four-dimensional (4D) microscopy, thereby making it easier to quantify the size and number of LDs compared to NR and other dyes [215].

The purification and proteomic analysis of *C. elegans* LDs revealed that the short-chain dehydrogenase DHS-3 that belongs to the HSD family was almost completely localized and highly enriched on the surface of the LDs [216]. The fluorescence signals detected in the DHS-3::GFP strain almost merged with the signals of the LDs detected by DIC microscopy or NR staining. Thus, DHS-3 can also be used as an LD-specific marker protein in nematodes. Therefore, DHS-3::GFP has been used for the forward genetic screening and identification of mutant strains with an altered LD morphology. Using DHS-3 as a LD marker, the deletion of MDT-28/PLIN-1 or C27H5.2 was proved to cause the aggregation of LDs in *C. elegans* [217], and 78 genes associated with significant changes in the morphology of the LDs were also identified by a whole-genome RNAi screen [218]. DHS-3, MDT-28, and some other LD-associated proteins have been verified as LD residents [219], whose fusion with GFP can more clearly characterize the size, number, and distribution pattern of LDs and help reveal the regulatory mechanisms of the storage of fat.

### 5.3. Biochemical and Chemical Assay

Although the above staining-based methods can easily observe and characterize the lipid stores of nematodes, they do not estimate the absolute content of specific lipids. Therefore, chemical and biochemical assays are widely used to analyze the composition and absolute levels of the lipids in *C. elegans*.

#### 5.3.1. TG Quantification in *C. elegans*

The triglyceride level is usually determined as an endpoint to indicate the accumulation of fat. In this method, lipases hydrolyze TGs in nematode homogenates to release glycerol and FFAs [220]. Then, the glycerol is phosphorylated by glycerol kinase (GK) to produce glycerol-3-phosphate, which is further oxidized to dihydroxyacetone phosphate with the production of hydrogen peroxide (H_2_O_2_). Finally, H_2_O_2_ is measured by the Trinder reaction to calculate the TGs content in the sample. Relying on the extraction of the total TGs from the whole worm population, the TG levels have been normalized to either the total protein levels or phospholipid levels [221].

The enzymatic determination of the TG levels can be used to study changes in the total TG content due to genetic perturbations, dietary changes, or pharmacological disturbances. Recently, using the enzymatic measurements of TAGs, the fat-reducing effects of some natural product, active ingredients, such as polysaccharides from *Auricularia auricula* [222], *Volvariella volvacea* [223], bitter melon [224] as well as phenolics from strawberry and raspberry [225], have been identified in *C. elegans*. In another study, Crawford and co-workers applied the enzymatic measurements of TGs and found that the lipid content of the diet could modify the lipid accumulation caused by methylmercury [226].

The quantification of the total TGs by enzymatic assays in *C. elegans* is less influenced by external factors and is easy to operate, with a good stability and reproducibility. However, it is not feasible for high-throughput screening or determining the spatial distribution of fat in different tissues or compartments. In addition, large amounts (thousands) of *C. elegans* are required to obtain enough lipids, which makes the culture and homogenization steps of the worms very labor-intensive.

#### 5.3.2. TLC

TLC is widely applied to separate and quantify lipid species, such as fatty acids, glycerophospholipids, ceramides, TGs, steroids, and sphingolipids, as its setup is least dependent on specialized equipment and its chromatographic procedure is straightforward.

In order to analyze the lipids extracted from *C. elegans*, they are loaded near one edge of the glass plate coated with the adsorbent silica and then dipped into a chamber containing a mixture of solvents. The mixture of solvents moves along the plate together with the lipids sample by a capillary action. The lipid mixture is separated into a series of spots due to the different solubility of the components in the solvents and the difference in the ability of the adsorbent to adsorb them. Finally, the fractionated lipids can be detected by appropriate methods such as staining the lipids by spraying plates with dyes or detecting lipid spots by iodine vapor. Lipid species can be compared to the standards for identification and quantification purposes [221].

The operation and equipment of TLC are simple and low-cost. In addition, TLC has the advantages of an easy color development and high sensitivity, but it is not suitable for the separation and analysis of lipids in small doses of biological samples, so an enormous number of nematodes are required for the experiment.

#### 5.3.3. GC

A biochemical quantification is useful for the detection of the glycerol moiety of the TGs, but it does not determine the fatty acid chain lengths, composition, and saturation levels. In contrast, individual GC or GC in combination with other methods enables the sensitive detection and relative quantification of fatty acids in total nematode lipids. In general, the fatty acid methyl ester (FAME) derivatization of whole worms has been applied to analyze the individual fatty acid abundances of all lipid species [227]. However, to analyze the abundance of fatty acids within specific lipids and calculate the neutral lipid stores, the separation and enrichment of one type of lipid can be achieved by a solid phase extraction (SPE) pretreatment and preparative chromatography such as TLC [53,228], high performance liquid chromatography (HPLC) [229], and ultra-performance liquid chromatography (UPLC) [230]. The chromatographically separated lipids are subsequently subjected to FAME derivatization. The FAMEs obtained by the above methods transition to the gas phase after heating, which are separated by GC and subsequently detected on a GC-coupled mass spectrometer [227].

To date, many studies have used GC-based approaches to profile the total fatty acids presenting in the lipid metabolism and its regulation, the effects of an exposure to different chemicals, and the metabolic changes induced by the genetic interference in nematodes [33]. Watts and Browse screened and isolated *C. elegans* mutants lacking PUFA biosynthesis by the GC analysis of the composition of fatty acids, while revealing the substrate preferences of desaturase/elongase enzymes [51]. Later, via GC coupled to a flame ionization detector (GC-FID), the genetic regulation of the composition of UFAs in the nematodes were announced [53]. One of the more recent investigations was performed by Wei et al., where GC-mass spectrometry (GC-MS) was used to test the FA contents and the results demonstrated the disruption of perfluorooctane sulfonate on the FA metabolism in *C. elegans* [231]. Furthermore, a novel method using a combination of electron impact ionization (EI) and collision-induced dissociation coupled to GC/MS (GC-EI-MS) has been applied to accurately quantify the anti-aging lipid compound oleoylethanolamine (OEA) in nematodes, providing an excellent method for measuring the OEA dynamics under different genetic and environmental perturbations [232].

Collectively, GC-FID and GC-MS are the reliable and highly sensitive methods most widely used for the analysis of FAs and sterols, while GC-EI-MS has been evaluated as an additional tool for the characterization of lipid compounds such as OEA, N-acylethanolamines (NAEs) [233], and ascarosides [234] in *C. elegans*.

#### 5.3.4. Mass Spectrometry-Based Methods

Nowadays, lipidomics is mainly based on MS. The shotgun method has been mostly used for proteomic analysis in *C. elegans* [235,236], while it has also been gradually applied for lipid studies in nematodes in the last decade.

In shotgun lipidomics, the total lipid extracts of *C. elegans* are directly and stably infused into a mass spectrometer without a prior chromatographic separation. Subsequently, individual class of lipid molecules are identified and quantified by the methods relying on tandem mass spectrometry (MS/MS), such as precursor ion scanning, neutral loss scanning, and product ion scanning [237]. Lipid species are quantified by specific fragment ions in the MS^n^ spectra, primarily by ion trap MS, but also Quadrupole-Time of Flight (Q-ToF) instruments. Moreover, Fourier transform (FT) MS is used for the direct identification and quantification of the lipid species as it can distinguish the *bona fide* lipid peaks from chemical noise and enables the determination of the composition of lipid molecular species bypassing MS/MS [238].

Based on the shotgun mass spectra, the *C. elegans* lipid extracts were directly profiled to validate their data-dependent acquisition method and the pronounced perturbations in the abundance of lipid precursors were revealed [239]. In another study, the lipid extracts of *C. elegans* larvae at the dauer and L3 stages were screened with the shotgun lipidomics approach, from which a new class of specifically enriched lysomaradolipids was recognized [240]. Although the application of shotgun in nematode lipid studies is unusual, it is agreed that shotgun lipidomics will be widely used due to its attractive advantages such as the simplicity of the process, high resolution, and automation of the sample-related operations. However, shotgun lipidomics cannot directly distinguish between the isomeric and isobaric species and ion suppression effects, which may limit its ability to detect low-abundance components.

As a powerful and sensitive analytical technique, liquid chromatography mass spectrometry (LC-MS) can systematically analyze the composition and content of the intact lipids of *C. elegans*. Unlike the shotgun approach, LC/MS requires the simultaneous separation of multiple lipid samples from nematodes by the liquid chromatographic column beforehand, and then the qualitative information is obtained from the lipid profile and chromatographic retention time for the structural identification of the lipid molecules. Subsequently, the lipids’ quantification relies on the intensity of the response, such as the peak height and peak area in lipid chromatography.

Typically, the intensity and retention variation in the Cortecs C18 column was found to be stable within a good range, and it showed an excellent performance for the lipid profiling of *C. elegans* compared to that of the conventional C8 column [241]. Using the Cortecs C18 column, a reversed-phase LC-MS (RP-LC-MS) was performed, showing that the improved combined extraction scheme for lipid and metal species allowed for the simultaneous monitoring and analysis of the lipidomes of the same nematode population, especially iron redox state homeostasis [242]. Through the LC-MS/MS method, Muthubharathi et al. found that a *sakazakii* infection made the nematode neurotransmitters and fatty acids abnormal, which was associated with the accumulation of LD [243]. In addition, LC-MS and multiple reaction monitoring (MRM) were also combined to fully characterize the composition and abundance of sphingolipids (SLs) under different developmental stages and growth conditions in nematodes. In this case, many culturing condition-dependent metabolic features of the *C. elegans* SL composition were spotted [244]. To sum up, the application of LC-MS-based lipidomics will be gradually increased, as it does not require the long isolation, derivatization, and quantification of FAs.

## 6. Summary and Perspectives

*C. elegans* is a well-established animal model that has been widely used in obesity studies. Although they lack specific organs and circulatory systems, about 500 of more than 20,000 genes in their genome can regulate adiposity. Both the genetic manipulability of nematodes and their homology with mammals in the genes related to the lipid metabolism are effective advantages for investigating the regulators of body fat, lipid regulation pathways, and metabolism-related disease mechanisms.

Furthermore, the continuous improvement of fixed staining and label-free methods for characterizing the accumulation of the lipids in nematodes has made it easier and more accurate to observe the distribution of lipids in tissues and measure the lipid levels quantitatively. As for the systematic measurement of complex lipids and the specific analysis of their composition, nematodes can be studied by lipidomics, a critical important branch of metabolomics. Herein, most analytical techniques such as chromatography, mass spectrometry, and their combinations are applied for the separation and analysis of lipids and their structural identification. However, the lack of appropriate databases and the inability of the existing techniques for a lipid analysis to simultaneously detect all the lipids in nematodes at once have limited the development of lipidomics in nematodes. Nevertheless, there is still plenty of scope to combine lipidomics with other metabolomics transcriptomics, and proteomics for nematode lipid analysis, and it is expected that more applications of lipidomics in nematodes will be available in the future.

Based on the superiority of nematodes as a model for lipid metabolism research and the maturity of the experimental methods for characterizing lipids, nematodes are increasingly applied in the screening and research of genes and their chemical compounds with lipid-altering functions. With the growing epidemic of obesity, it is crucial to maintain normal body lipid levels in the organism. In this context, determining the genes and chemical compounds regulating the lipid metabolism and further studying their mechanisms in depth can help maintain a balance of energy and the control of body fat. Specifically, an exposure to the chemical compounds evaluated as lipid-elevating should be minimized, while chemical compounds with lipid-lowering effects can be used in the diet or developed as lipid-lowering drugs if they have no other side effects on the organism. Meanwhile, the elucidation of the mechanisms of action of the relevant genes and chemical compounds that lower blood lipids will facilitate therapeutic interventions for obesity and related diseases. Unfortunately, it is not yet possible to extrapolate the effective doses of the compounds tested in *C. elegans* to humans and other animal models. Therefore, the significance of the doses used in nematodes needs to be examined in greater depth.

Using *C. elegans* as an excellent model for studying the lipid metabolism, more genes and substances with body fat regulatory functions will be screened and studied in the future. The clarification of the intricate signaling pathways in the lipid metabolism will also help to identify more instructive body fat regulators. The spatialization of the cellular compartments in *C. elegans* makes it possible to explore the lipid metabolism in the individual cells of the organism and assists in understanding the systemic mechanisms of how different cells interact to regulate the lipid and energy status throughout the body. The availability of more and more experimental techniques and tools will make lipid research more convenient and faster. As a brief example, GFP- or RNAi-based technologies allow studies related to the regulation of the whole-body lipid levels and energy status in nematodes to be feasible for the entire genome. It is expected that the regulation of the body fat levels will be increasingly emphasized in the coming years, and nematodes are an appreciated model to explore the regulatory logic of the control of body fat.

In summary, this review mainly focuses on lipid metabolic processes and the related pathways regulating the storage of lipids as well as the experimental methods for the characterization of lipids in *C. elegans*. As *C. elegans* models are increasingly used in to the field of research involved in metabolic homeostasis, such as aging, stress resistance, reproduction, and neurodegenerative diseases, the association between the balance of energy in these processes and adiposity is sure to become a focus of future research and be explored intensively.

## Figures and Tables

**Figure 1 ijms-24-01173-f001:**
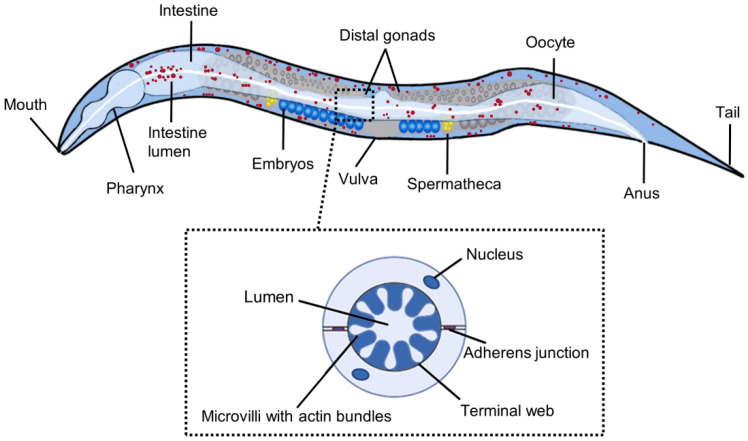
Anatomical diagram of an adult nematode.

**Figure 2 ijms-24-01173-f002:**
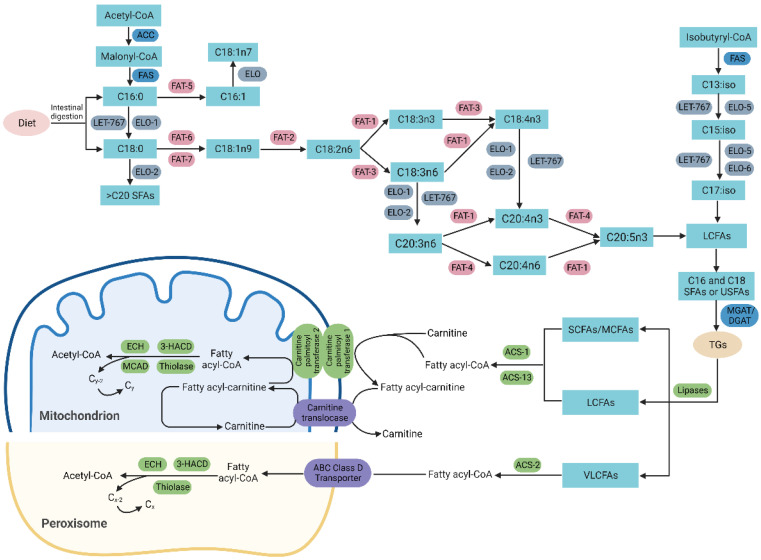
Synthesis and breakdown of fatty acids in *C. elegans*.

**Figure 3 ijms-24-01173-f003:**
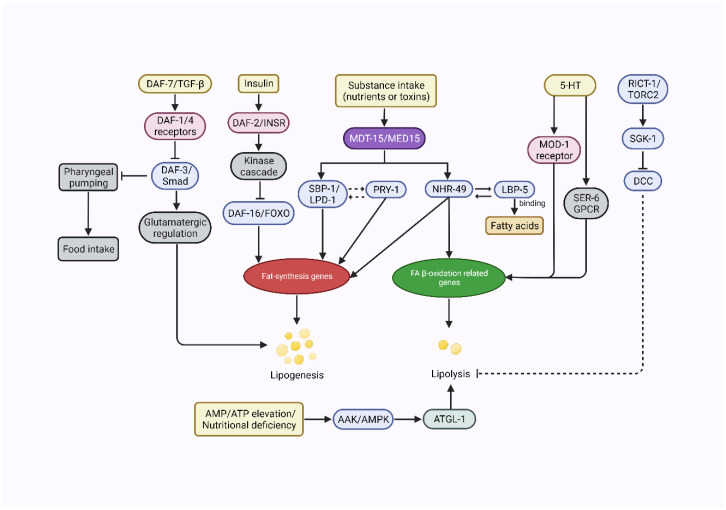
Proposed pathways regulating lipid levels in *C. elegans.* Abbreviations: TGF-β, transforming growth factor-β; INSR, insulin receptor; NHR, nuclear hormone receptor; GPCR, G protein-coupled receptor; DCC, dosage compensation complex; AMP, adenosine monophosphate; ATP, adenosine triphosphate.

**Figure 4 ijms-24-01173-f004:**
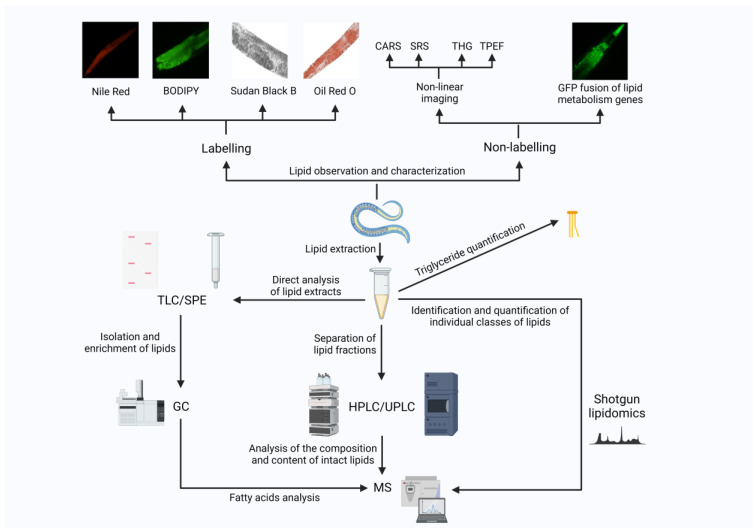
Overview of different strategies and common methods to analyze lipids and lipid deposits in *C. elegans* studies.

## Data Availability

Not applicable.

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
