# Peer review of "Application of Caenorhabditis elegans in Lipid Metabolism Research"

_ijms, 2023, doi:10.3390/ijms24021173_

Round 1

Reviewer 1 Report

To the authors, 

Manuscript Title: Application of Caenorhabditis elegans in lipid metabolism research

Comments: In this manuscript, An et al., have presented comprehensive review on the utility of Caenorhabditis elegans as a model for lipid metabolism research. The authors have provided a detailed analysis on lipid metabolic processes and regulatory pathways which can also be studied in C. elegans. In general, this review is straight forward and is worthy of publication. It is clearly evident that the authors have put considerable time and effort in writing this manuscript as it can be gathered from their writing. Listed below are areas of strength and few minor concerns I want to present for the authors consideration for revision in the manuscript.

Strengths:

1. The review is well written and easy to read through.

2    2. The authors have provided a comprehensive analysis on critical lipid regulatory pathways and how C. elegans can be used to study them as 70% of genes regulating lipid metabolism is homologous between C. elegans and humans. This is well done throughout the manuscript.

3   3. The authors have also provided a detailed overview of the different and current techniques used in the field of lipid research and how it can be used in model organisms such as C. elegans. This is the highlight of the manuscript as the authors submit clear arguments on the utility of using C. elegans in lipid research.

Minor Concerns:

1   1. The authors have mentioned a lot of genes to draw an example of gene regulations and manipulations conducted to study lipid metabolism. It would be great if the authors could include the gene names before the abbreviations.

2   2. The authors should use the complete name of organisms before including the abbreviated nomenclature (for example: the authors need to write the full name of S. maltophilia in the article).

3   3. I have concern regarding already published images as figures for this manuscript. I request the authors to provide justification as to why they opted to use published figures (Figure 2, 3 and 4) instead of making a new original figures for the manuscript. 

Author Response

Response to Reviewer 1 Comments

Thank you for your comments concerning our manuscript entitled “Application of Caenorhabditis elegans in lipid metabolism research” (ID: IJMS-2105325). These comments are all valuable and very helpful for revising and improving our paper. We have studied the comments carefully and have tried our best to revise the manuscript. In addition, the manuscript was checked thoroughly and also polished by a native speaker in writing.

Point 1: The authors have mentioned a lot of genes to draw an example of gene regulations and manipulations conducted to study lipid metabolism. It would be great if the authors could include the gene names before the abbreviations.

Response 1: Thank you for your comments, and we are grateful for the useful suggestion. We have prefixed the full gene name to all gene abbreviations that appear for the first time in the manuscript.

Point 2: The authors should use the complete name of organisms before including the abbreviated nomenclature (for example: the authors need to write the full name of S. maltophilia in the article).

Response 2: Thank you for your valuable advice. We checked the names of all organisms appearing in the manuscript. For names with abbreviations only, we added their full name in front as per your suggestion. For example, “S. maltophilia” was modified to “Stenotrophomonas maltophilia (S. maltophilia)”.

Point 3:  I have concern regarding already published images as figures for this manuscript. I request the authors to provide justification as to why they opted to use published figures (Figure 2, 3, and 4) instead of making new original figures for the manuscript.

Response 3: Thank you for your attention and valuable advice, and it is indeed sensible to make new original figures for our manuscript. Hence, based on the understanding of the relevant knowledge and literature, we try our best to create new original images for the manuscript, as shown in Figure 1, 3, and 4. Besides, Figure 2A (about lipogenesis) is originally a new one which is created before being submitted. Concerning lipolysis, Figure 2B in the original manuscript was innovative, comprehensive, and integrated, which made us keep it as a part of Figure 2. We hope that readers can read the article in conjunction with the images to deepen their understanding of C. elegans and lipid metabolism.

Reviewer 2 Report

In this review manuscript, the authors have described C. elegans as a model of lipid metabolism. I believe incorporating the following suggestions will significantly improve the quality and readability of this manuscript.

The authors have briefly detailed the advantages of using C. elegans as a model to study lipid metabolism; they should also discuss the disadvantages of C. elegans as a model to study lipid metabolism.

Although the authors described the general physiology of lipid reserves in C. elegans, a graphical illustration of anatomical structures of C. elegans lipid reservoirs will be helpful to experts from non-c. elegans field. This illustration can be accompanied by lipid uptake, transport, synthesis, and catabolism. 

For better understanding, Fig1 and 2 can be merged into a single figure.

Recently, lipid metabolism in C. elegans is also correlated with pathogenesis, e.g., pseudomonas perception and infection. This is also a unique advantage of C. elegans as a model, and authors should detail its role in identifying novel mechanisms of regulation of lipid metabolism, which are seldom seen in other models.

For example, please see 

-Anderson SM, et al., (2019) The fatty acid oleate is required for innate immune activation and pathogen defense in Caenorhabditis elegans. PLoS Pathog 15(6): e1007893. https://doi.org/10.1371/journal.ppat.1007893

-Nhan et al., Redirection of SKN-1 abates the negative metabolic outcomes of a perceived pathogen infection, 2019 PNAS

C. elegans is an established model for studying aging, and recent evidence demonstrated the link between lipid levels and longevity. C. elegans is uniquely advantageous for examining this interaction. Authors should consider introducing a subsection about this link.

Fatty acid supplementation has been shown to regulate several different biological processes in C. elegans. Authors should elaborate on this unique aspect of C. elegans lipid biology.

- e.g., Dana A. Lynn et al., Omega-3 and -6 fatty acids allocate somatic and germline lipids to ensure fitness during nutrient and oxidative stress in Caenorhabditis elegans

-Deline ML, et al., Dietary supplementation of polyunsaturated fatty acids in Caenorhabditis elegans. J Vis Exp. 2013

-Qi W, et al., The ω-3 fatty acid α-linolenic acid extends Caenorhabditis elegans lifespan via NHR-49/PPARα and oxidation to oxylipins. Aging Cell. 2017

-A Ranawade et al., PRY-1/Axin signaling regulates lipid metabolism in Caenorhabditis elegans. PLoS one, 2018 demonstrated the involvement of Wnt signaling in lipid metabolism.

And several other papers.

In section 4. "Conserved fat metabolism signaling pathways in C. elegans" authors should include other conserved but less known signaling pathways implicated in regulating lipid metabolism in C. elegans. For example, A Ranawade et al., PRY-1/Axin signaling regulates lipid metabolism in Caenorhabditis elegans. PLoS one, 2018 demonstrated the involvement of Wnt signaling in lipid metabolism.

Other discoveries that can be incorporated in this review to demonstrate the utility of the C. elegans model in lipid biology

  • Lipase Action and a Link to Autophagy
  • How tissue-specific functions can be investigated in C. elegans e.g., Muscle-Specific Lipid Hydrolysis Prolongs Lifespan through Global Lipidomic Remodeling, Schmeisser et al.,

Although the Section 5. "Experimental tools for the studies of fat metabolism in C. elegans" provides a detailed overview of the tools available to investigate lipid biology in C. elegans; this section is far too long. The authors should succinctly introduce the technology and highlight the key findings and advantages of C. elegans.

The authors should enlist and discuss the Future directions of C. elegans lipid biology research.

Author Response

Response to Reviewer 2 Comments

Thank you for your comments concerning our manuscript entitled “Application of Caenorhabditis elegans in lipid metabolism research” (ID: IJMS-2105325). These comments are all valuable and very helpful for revising and improving our paper. We have studied the comments carefully and have tried our best to revise the manuscript. In addition, the manuscript was checked thoroughly and also polished by a native speaker in writing.

Point 1: The authors have briefly detailed the advantages of using C. elegans as a model to study lipid metabolism; they should also discuss the disadvantages of C. elegans as a model to study lipid metabolism.

Response 1: Thank you for your comments, and we are grateful for the useful suggestion. As everything has two sides, the disadvantages of C. elegans as the model to study lipid metabolism cannot be ignored. In order to give the reader a more comprehensive understanding of the application of nematodes in lipid research, we have presented the disadvantages of nematodes as a model for studying lipid metabolism in the last paragraph of section 2 (2. Advantages and disadvantages of C. elegans as a model for studying fat metabolism) of the manuscript.

Point 2: Although the authors described the general physiology of lipid reserves in C. elegans, a graphical illustration of anatomical structures of C. elegans lipid reservoirs will be helpful to experts from non-c. elegans field. This illustration can be accompanied by lipid uptake, transport, synthesis, and catabolism. For better understanding, Fig1 and 2 can be merged into a single figure.

Response 2: Thank you for your specific and instructive suggestions. We apologize for any inconvenience that may be caused to non-C. elegans experts by neglecting to illustrate the anatomy of nematodes. Therefore, we supplemented Figure 1, an anatomical diagram of an adult C. elegans. In the figure, we pointed out the main organs of nematode feeding, fat storage and mobilization. However, we failed to put it together with the diagram of lipogenesis and lipolysis. This is because the synthesis and decomposition of fat are microscopic processes that occur in many cells, and drawing them together would prevent many of the details and specific steps of lipogenesis and lipolysis from being encapsulated and highlighted. To make it as easy as possible for the reader to understand, as shown in Figure 2, we have combined the diagrams of lipogenesis and lipolysis according to your suggestion.

Point 3: Recently, lipid metabolism in C. elegans is also correlated with pathogenesis, e.g., pseudomonas perception and infection. This is also a unique advantage of C. elegans as a model, and authors should detail its role in identifying novel mechanisms of regulation of lipid metabolism, which are seldom seen in other models.

Response 3: Thank you for your valuable advice. In the fourth paragraph of section 2 of the manuscript (Page 2-3), we have added the association between nematode lipid metabolism and pathogenic mechanisms, which is undoubtedly very rare in other models and is another advantage of nematodes for lipid studies.

Point 4: C. elegans is an established model for studying aging, and recent evidence demonstrated the link between lipid levels and longevity. C. elegans is uniquely advantageous for examining this interaction. Authors should consider introducing a subsection about this link.

Response 4: We are grateful for the useful suggestion. Nematodes have been often used as a wonderful model for aging-related studies. Obesity has also been extensively studied in C. elegans because of similar cellular and metabolic disorders to aging. Therefore, we have added a discussion of the association between lipid levels and longevity in C. elegans in the fifth paragraph of section 2 of the manuscript (Page 3).

Point 5: Fatty acid supplementation has been shown to regulate several different biological processes in C. elegans. Authors should elaborate on this unique aspect of C. elegans lipid biology.

Response 5: Thank you for your professional guidance and suggestions. In recent studies, fatty acid supplementation has become a common means of altering the fatty acid composition of nematodes and has been shown to modulate nematode biological processes. In view of this unique aspect, we have added subsection “3.4 Fatty acid supplementation” to section 3 of the manuscript (Page 7-8).

Point 6: In section 4. "Conserved fat metabolism signaling pathways in C. elegans" authors should include other conserved but less known signaling pathways implicated in regulating lipid metabolism in C. elegans. For example, A Ranawade et al., PRY-1/Axin signaling regulates lipid metabolism in Caenorhabditis elegans. PLoS one, 2018 demonstrated the involvement of Wnt signaling in lipid metabolism.

Response 6: Thank you for your valuable comments and suggestions. According to your advice, we have added a subsection "4.7 PRY-1/Axin signaling " to section 4 of the manuscript (Page 18-19), which provides details on the unusual lipid-related pathway that has been less studied and mentioned in C. elegans.

Point 7: Other discoveries that can be incorporated in this review to demonstrate the utility of the C. elegans model in lipid biology

  • Lipase Action and a Link to Autophagy
  • How tissue-specific functions can be investigated in C. elegans e.g., Muscle-Specific Lipid Hydrolysis Prolongs Lifespan through Global Lipidomic Remodeling, Schmeisser et al.,

Response 7: We appreciate the effort of the reviewer to review our manuscript and the accompanying valuable suggestions. The nematode model is practical for the study of lipid biology. According to your suggestion, we have added relevant content about the relationship between lipase and autophagy in the third paragraph of subsection 3.3 in the manuscript (Page 6-7). The study of tissue-specific function in C. elegans was also added in the fifth paragraph of section 2 of the manuscript (Page 3).

Point 8: Although the Section 5. "Experimental tools for the studies of fat metabolism in C. elegans" provides a detailed overview of the tools available to investigate lipid biology in C. elegans; this section is far too long. The authors should succinctly introduce the technology and highlight the key findings and advantages of C. elegans.

Response 8: Thank you for underlining this deficiency. In order to make the application and advantages of nematodes in lipid research techniques clearer and more prominent, unnecessary and insufficiently relevant contents in section 5 of the manuscript were deleted and some experimental tools were also summarized into concise sentences.

Point 9: The authors should enlist and discuss the Future directions of C. elegans lipid biology research.

Response 9: Thank you for the suggestion. We have included a relevant discussion on the future direction of nematode lipid biology research in the fourth paragraph of section 6 of the manuscript (Page 27-28).

Round 2

Reviewer 2 Report

I am happy that the authors have incorporated all the suggested changes; however, although these add value to the review, they need to be presented coherently. For example, the newly added information about pathogenesis and aging appears to be stand-alone paragraphs and does not highlight the strength of the C. elegans model in investigating lipid biology. In section 2. the authors should make a coherent and succinct case for the advantages of C. elegans as a model to study lipid metabolism. The present manuscript appears to be just a collection of information. 

Figure 1. Authors can include lipid deposits (highlighted in red color) in the images for better representation.

Figure 2 Authors used two different types of pathway representations in A and B. I suggest Author merge these pathways (using one style of presentation)

Figure 3. The authors incorrectly used "proposed pathways"; these are well-established evolutionarily conserved pathways. 
The newly added pathways are not represented in the Figure 3.
Authors should make necessary changes to reflect this fact. 

Section 3.4 doesn't fit well in section 3.

line 974: "The spatialization of cellular compartments in C. elegans makes lipid  biology studies applicable to the whole genome." It is not clear from this sentence how cellular organization influences the whole genome. Do the authors mean systemic mechanisms to study how different cellular types interact with each other to regulate lipid and energy status in the whole body?

Author Response

Response to Reviewer 2 Comments

Thank you for your comments concerning our manuscript entitled “Application of Caenorhabditis elegans in lipid metabolism research” (ID: IJMS-2105325). These comments are all valuable and very helpful for revising and improving our paper. We have studied the comments carefully and have tried our best to revise the manuscript. In addition, the manuscript was checked thoroughly and also polished by a native speaker in writing.

Point 1: In section 2. the authors should make a coherent and succinct case for the advantages of C. elegans as a model to study lipid metabolism. The present manuscript appears to be just a collection of information. 

Response 1: Thank you for your comments, and we are grateful for the useful suggestion. We apologize for not highlighting the advantages of the nematode model for lipid studies in the newly added information on pathogenesis and aging in section 2. In addition to the characteristics of the nematode itself, we believe that the discovery of the correlation of lipid metabolism with other physiological processes in C. elegans is undoubtedly its strength in lipid biology research. Therefore, we have revised the content of the fourth and fifth paragraphs of section 2 (Page 3-4) so that the advantages of nematodes as a model for studying lipid metabolism research are presented coherently and concisely.

Point 2: Figure 1. Authors can include lipid deposits (highlighted in red color) in the images for better representation.

Response 2: Thank you for your valuable advice. Although nematode has no adipose tissue, it stores fat mainly in lipid droplets in intestinal and subcutaneous tissue cells. Therefore, to better demonstrate and express the distribution of lipids in nematodes, we have added and highlighted the lipid droplets in red color in Figure 1(Page 5).

Point 3: Figure 2 Authors used two different types of pathway representations in A and B. I suggest Author merge these pathways (using one style of presentation).

Response 3: Thank you for your attention to the figure in the manuscript, and we apologize for using two different types of pathway representations. Based on our understanding of the relevant knowledge and literature, we try our best to create a new image for the manuscript, as shown in Figure 2 (Page 10). In the new image, we merged the pathways of synthesis and breakdown of fatty acids and represented them using the same presentation style. We hope that readers can read the article in conjunction with the images to deepen their understanding of C. elegans and lipid metabolism.

Point 4: Figure 3. The authors incorrectly used "proposed pathways"; these are well-established evolutionarily conserved pathways.

The newly added pathways are not represented in Figure 3.

Authors should make necessary changes to reflect this fact.

Response 4: Thank you for your professional guidance and suggestions. We apologize for misrepresenting conserved fat metabolism signaling pathways in C. elegans. To compensate for this mistake, we modified the name of Figure 3 to "Major evolutionarily conserved pathways regulating lipid levels in C. elegans". In addition, we represented the newly added PRY-1/Axin signaling in Figure 3 (Page 12), showing the facilitative effect of PRY-1 on fat-synthesis genes and its interaction with SBP-1, which remains to be determined and further studied.

Point 5: Section 3.4 doesn't fit well in section 3.

Response 5: Thank you for your valuable advice. Considering that nematodes are superior models for studying fatty acid function, they are able to independently synthesize a range of omega-3 and omega-6 fatty acids and alter fatty acid composition through dietary supplementation. The advantage of nematodes in fatty acid supplementation is summarized in the third paragraph of Section 2 of the manuscript (Page 2-3). It is hoped that through this passage the reader will gain a better understanding of the application of nematodes to this unique aspect of lipid biology.

Point 6: line 974: "The spatialization of cellular compartments in C. elegans makes lipid biology studies applicable to the whole genome." It is not clear from this sentence how cellular organization influences the whole genome. Do the authors mean systemic mechanisms to study how different cellular types interact with each other to regulate lipid and energy status in the whole body?

Response 6: We appreciate the effort of the reviewer to review our manuscript and the accompanying valuable suggestion. We apologize for not clearly expressing the concept of this sentence. With this statement, we had intended to express that the spatialization of cellular compartments in C. elegans makes it possible to explore lipid metabolism in individual cells of the organism and assists in understanding the systemic mechanisms of how different cell types interact to regulate lipid and energy status throughout the body. Furthermore, GFP- or RNAi-based technologies allow studies related to the regulation of whole-body lipid levels and energy status in nematodes to be feasible for the entire genome. We have made modest changes to this paragraph in the manuscript (Page 30-31) for better understanding by reviewers and readers.

Round 3

Reviewer 2 Report

I am satisfied with the revised manuscript and support publication of this review article.